# Compact meta-differentiator for achieving isotropically high-contrast ultrasonic imaging

Yurou Jia[1,2], Suying Zhang[1], Xuan Zhang[1], Houyou Long [1] ✉, Caibin Xu[3], Yechao Bai [4], Ying Cheng [1,5] ✉, Dajian Wu [6], Mingxi Deng[3], Cheng-Wei Qiu[2] & Xiaojun Liu [1,5] ✉

Ultrasonic imaging is crucial in the fields of biomedical engineering for its deep penetration capabilities and non-ionizing nature. However, traditional techniques heavily rely on impedance differences within objects, resulting in poor contrast when imaging acoustically transparent targets. Here, we propose a compact spatial differentiator for underwater isotropic edge-enhanced imaging, which enhances the imaging contrast without the need for contrast agents or external physical fields. This design incorporates an amplitude meta-grating for linear transmission along the radial direction, combined with a phase meta-grating that utilizes focus and spiral phases with a first-order topological charge. Through theoretical analysis, numerical simulations, and experimental validation, we substantiate the effectiveness of our technique in distinguishing amplitude objects with isotropic edge enhancements. Importantly, this method also enables the accurate detection of both phase objects and artificial biological models. This breakthrough creates new opportunities for applications in medical diagnosis and nondestructive testing.

Ultrasound imaging, renowned for its ability to visualize and analyze morphological features based on the contrast in mechanical properties of objects, plays a pivotal role in various applications, including medical diagnosis, acoustic microscopy, and nondestructive testing[1–5]. Particularly, in the field of biomedicine, ultrasound imaging offers advantages such as deep penetration, non-ionizing nature, minimal invasiveness, and low cost. It has proven valuable in assessing the biomechanical properties of cells and tissues, enabling early-stage diagnosis and treatment of diseased or cancerous tissue[6–13]. However, conventional ultrasonic imaging techniques heavily rely on impedance differences within the imaged objects. Higher impedance differences lead to clearer contrast, while lower differences limit the intrinsic

contrast between soft tissues with similar acoustic responses[9,14]. This limitation can significantly impact the ability to discriminate between different types of tissues, especially in neuroimaging and cardiovascular examinations. To address this limitation, the specialized ultrasound contrast agents have been introduced[15–19]. For example, air microbubbles with strong acoustic scattering characteristics are injected into the bloodstream as ultrasound contrast agents to provide sufficient impedance distinction, enhancing the contrast and visualization of targeted tissues[15–17]. Additionally, magnetic contrast agents are proposed for magneto-motive ultrasound imaging, which utilizes ultrasonic technology to monitor the mechanical displacements within tissues labeled by superparamagnetic nanoparticles in response to an

[1]Department of Physics, MOE Key Laboratory of Modern Acoustics, Collaborative Innovation Center of Advanced Microstructures, Nanjing University, Nanjing 210093, China. [2]Department of Electrical and Computer Engineering, National University of Singapore, Singapore 117583, Singapore. [3]College of Aerospace Engineering, Chongqing University, Chongqing 400044, China. [4]School of Electronic Science and Engineering, Nanjing University, Nanjing 210023, China. [5]State Key Laboratory of Acoustics, Institute of Acoustics, Chinese Academy of Sciences, Beijing 100190, China. [6]Jiangsu Key Lab on Opto-Electronic Technology, School of Physics and Technology, Nanjing Normal University, Nanjing 210023, China. ✉e-mail: longhouyou@nju.edu.cn; chengying@nju.edu.cn; liuxiaojun@nju.edu.cn

external magnetic field, leading to reconstructed images with improved contrast[18,19]. Moreover, exempt from the aid of contrast agents, photoacoustic imaging has been developed. It detects ultrasonic waves induced from superficial biological tissues irradiated by laser pulses due to thermoelastic expansion, providing informative optical contrast of tissues with low acoustic impedance difference but distinct optical absorption coefficient[20–22].

Current approaches primarily rely on contrast agents or external physical fields to enhance acoustic contrast. However, these techniques are expensive and have limitations, including the risk of allergic reactions, inapplicability in certain medical conditions, excretion from the body, and limited tissue penetrations. It is necessary to develop new methodologies for facilitating high-contrast acoustic imaging applications in the fields of biomedicine and engineering. On the other hand, in the field of optics, groundbreaking phase contrast methods like Zernike phase contrast[23] and differential interference contrast[24] have significantly improved optical contrast by enhancing the visibility of edges in amplitude and phase objects against the background medium[25–28]. Recent advancements in optical computational metamaterials, such as metasurfaces and photonic crystals, have demonstrated exceptional capabilities in edge-enhanced imaging through the utilization of spatial differentiation[29–34]. These advancements provide a crucial tool for identifying and highlighting boundaries between different regions of objects. This facilitates subsequent stages of pattern recognition by extracting important visual cues necessary for analyzing and interpreting images, which have significant applications in biological imaging, fingerprint identification, and industrial inspection[35–40]. Inspired by the advances in optics, researchers have also investigated acoustic computational metamaterials for differentiation operation[41–46]. This includes the use of metasurfaces based on the Fourier approach or phononic crystals employing the Green's function method[41–44]. However, despite their potential benefits in improved acoustic contrast, these methods have certain limitations that hinder their extensive applications. Firstly, most current demonstrations of spatial differentiation are limited to one-dimensional scenarios, resulting in anisotropic edge enhancements that are suboptimal for imaging applications[41,45,46]. Additionally, the use of multigroup metasurfaces and phononic crystals within waveguide in bulky configurations is incompatible with compact integrated systems[42–44]. Lastly, the current research primarily focuses on the edge detection of amplitude objects, with limited exploration of phase object detection. Consequently, there is a significant demand for a compact design that can achieve isotropic edge enhancement of both amplitude and phase objects through two-dimensional (2D) spatial differentiation.

In this paper, we propose a compact meta-differentiator based on acoustic composite meta-gratings that enables 2D isotropic edge enhancements for high-contrast ultrasonic imaging. Our meta-differentiator incorporates an amplitude meta-grating for linear transmission along the radial direction, with a phase meta-grating composed of focus and spiral phases carrying a first-order topological charge. This configuration allows for direct manipulation of the wavevector distribution, enhancing higher values while maintaining a central zero in the wavevector space to achieve 2D spatial differentiation. The processed imaging of amplitude objects exhibits distinct isotropic edge contrast enhancements, which are validated through theoretical analysis, numerical simulation, and experimental verification. Furthermore, we showcase the accurate and reliable isotropic edge-enhanced detection of both phase objects and artificial biological models resulting from the 2D spatial differentiation operation.

## Results

### Design of 2D spatial differentiator

Firstly, we discuss the principle of using an acoustic metasurface to achieve isotropic spatial differentiation. Figure 1a illustrates the arrangement of the input image, acoustic metasurface, and output image at different planes: $z = -f$, $z = 0$ and $z = f$, respectively. When an

input image with an acoustic field $p_1(x_1, y_1)$ passes through the acoustic metasurface with an aperture function $t_{\text{meta}}(x', y')$, the output image $p_2(x_2, y_2)$ can be calculated using the Rayleigh-Sommerfeld diffraction integral:

$$p_2(x_2,y_2) = -\frac{k^2}{4\pi^2}\iint\left[\iint p_1(x_1,y_1)\frac{e^{ikr_1}}{r_1}\,\mathrm{d}x_1\mathrm{d}y_1\right]t_{\text{meta}}(x',y')\frac{e^{ikr_2}}{r_2}\,\mathrm{d}x'\,\mathrm{d}y', \quad (1)$$

where the distance terms $r_{1,2} = \sqrt{(x' - x_{1,2})^2 + (y' - y_{1,2})^2 + f^2}$. By approximating Eq. (1) under the paraxial approximation, we can simplify and transform it into the following expression:

$$p_2(x_2,y_2) = -\frac{k^2}{4\pi^2}\frac{e^{ik\left(2f+\frac{x_2^2+y_2^2}{2f}\right)}}{f^2}\iint\left\{\iint p_1(x_1,y_1)e^{ik\frac{x_1^2+y_1^2}{2f}}\mathrm{d}x_1\mathrm{d}y_1\right\}$$
$$t_{\text{meta}}(x',y')e^{ik\frac{x'^2+y'^2}{f}}e^{-ik\frac{(x_1+x_2)x'+(y_1+y_2)y'}{f}}\mathrm{d}x'\,\mathrm{d}y'. \quad (2)$$

To tailor the spatial Fourier spectrum of the input image directly, the redundant phase term $e^{ik\frac{x'^2+y'^2}{f}}$ in Eq. (2) should be eliminated. For this purpose, we design the aperture function $t_{\text{meta}}(x', y')$ as follows:

$$t_{\text{meta}}(x',y') = e^{-i2k\left(\sqrt{x'^2+y'^2+f^2}-f\right)}t_{\text{tran}}(x',y'). \quad (3)$$

Here, the specific phase factor $e^{-i2k(\sqrt{x'^2+y'^2+f^2}-f)}$, known as the focus phase, can be approximated as $e^{-ik\frac{x'^2+y'^2}{f}}$ under the paraxial approximation. This phase factor is introduced to eliminate the redundant phase term and enable direct manipulation of the spatial Fourier spectrum (see Supplementary Note 1 for details). Additionally, $t_{\text{tran}}(x', y')$ represents the transfer function associated with mathematical operations in the spatial Fourier domain, which plays a crucial role in modulating spatial spectrum responses. By substituting Eq. (3), in an approximate form, into Eq. (2), the output image $p_2(x_2, y_2)$ can be reformulated as follows:

$$p_2(x_2,y_2) = -\frac{k^2}{4\pi^2 f^2}e^{ik\left(2f+\frac{x_2^2+y_2^2}{2f}\right)}\left\{p_1(-x_2,-y_2)e^{ik\frac{x_2^2+y_2^2}{2f}}\right\}\otimes\mathscr{F}\{t_{\text{tran}}(x',y')\}_{(k_x,k_y)}, \quad (4)$$

where the symbol $\otimes$ denotes the 2D convolution operation, and $\mathscr{F}\{t_{\text{tran}}(x',y')\}_{(k_x,k_y)}$ represents the Fourier transform of $t_{\text{tran}}(x', y')$ at the spatial wavevectors $k_x = \frac{k}{f}x_2$ and $k_y = \frac{k}{f}y_2$ along the $x$- and $y$-axes, respectively. From Eq. (4), the output image can be obtained by convolving the inverse input image with the mathematical operation $t_{\text{tran}}(x', y')$ in the spatial frequency domain (Supplementary Note 2)[36,37]. For example, in Fig. 1a, the conventional imaging system with $t_{\text{tran}}(x', y') = 1$ is capable of reconstructing the complete image of objects at the output plane, as shown in the insets illustrating the phase and amplitude profiles imparted onto the acoustic metasurface. Nevertheless, this imaging technique heavily relies on the intensity distribution of the acoustic field while neglecting the crucial phase information[14].

To extract the edge information of objects for high contrast imaging, the transfer function $t_{\text{tran}}(x', y')$ needs to be redesigned for spatial differentiation operations[38]. So far, extensive research has been conducted on the first-order (1st-order) differential operator $\left(\left|\frac{\mathrm{d}p_1(x_1)}{\mathrm{d}x_1}\right|\ \text{or}\ \left|\frac{\mathrm{d}p_1(y_1)}{\mathrm{d}y_1}\right|\right)$, and the corresponding transfer functions in the Fourier domain are $t_{\text{tran}}(x') \propto ix'$ or $t_{\text{tran}}(y') \propto iy'$ along the $x$- or $y$-axis, respectively[42,43]. By considering the linearity of Fourier transform, the 2D spatial

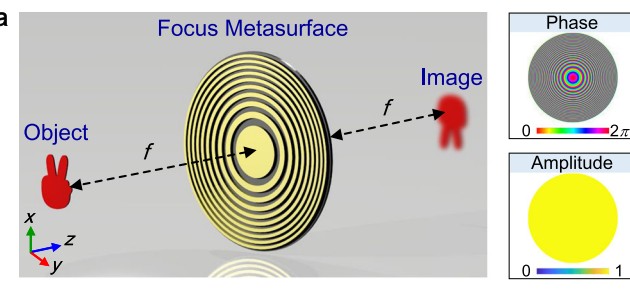

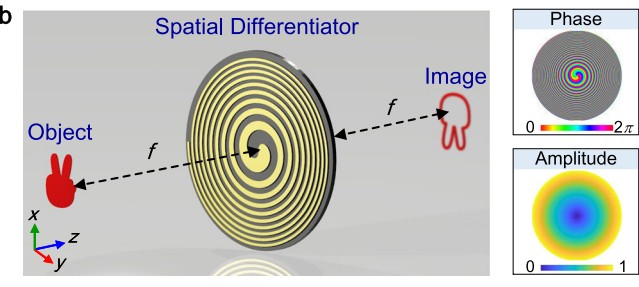

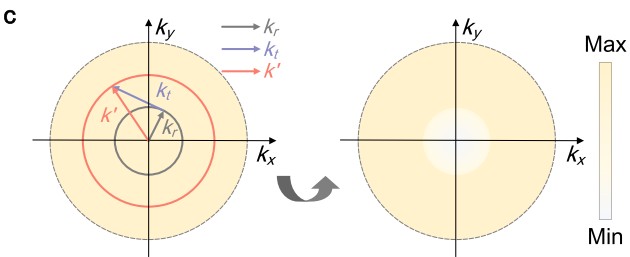

**Fig. 1 | Working principle of a 2D spatial differentiator for isotropic edge-enhanced imaging.** Schematic of the imaging process in (**a**) a conventional imaging system and (**b**) an edge-enhanced imaging system. The input image, acoustic metasurface, and the output image are positioned at different planes: $z = -f$, $z = 0$ and $z = f$, respectively. The insets in (**a**) and (**b**) show the amplitude and phase profiles imparted onto the acoustic metasurface. **c** Illustration of the spatial wavevector redistribution mechanism for the 2D spatial differentiation operation.

differentiation $\left( \left| \dfrac{\mathrm{d}p_1(x_1)}{\mathrm{d}x_1} + i\dfrac{\mathrm{d}p_1(y_1)}{\mathrm{d}y_1} \right| \right)$ can be obtained by superimposing both the $x$- and $y$-axial transfer functions, yielding:

$$t_{\mathrm{tran}}(x',y') = t_{\mathrm{tran}}(x') + it_{\mathrm{tran}}(y') \propto i(x' + iy'). \tag{5}$$

By integrating Eq. (5) into Eq. (3), the $t_{\mathrm{meta}}(x',y')$ of the acoustic metasurface can be expressed as:

$$t_{\mathrm{meta}}(x',y') = e^{-i2k\left(\sqrt{x'^2+y'^2+f^2}-f\right)} i(x'+iy') = e^{-i2k\left(\sqrt{r^2+f^2}-f\right)} re^{i\theta}, \tag{6}$$

where $r$ and $\theta$ represent the radial and azimuthal coordinates at the metasurface plane, respectively. As illustrated in Fig. 1b, the 2D differentiation operations yield isotropic edge enhancements of objects. The insets in the figure show the amplitude and phase distributions applied to the 2D spatial differentiator. The amplitude modulator $A = |r|$ exhibits linear transmission along the radial direction, while the phase modulator $\varphi = -2k\left(\sqrt{r^2+f^2}-f\right)+\theta$ combines the focus phase with a spiral phase carrying a topological charge of 1. In comparison to the Fourier approach based on multigroup metasurfaces[42], the meta-differentiator proposed here offers the advantage of a more compact volume through its single-layer design. Figure 1c presents a schematic illustration of the spatial frequency redistribution mechanism in Fourier

space. According to the generalized Snell's law, the focus phase generates a wavevector $k_r$ along the radial direction, while the spiral phase introduces an additional wavevector $k_t$ along the tangential direction (see Supplementary Note 3). The vector sum of $k_r$ and $k_t$ contributes to a higher wavevector $k'$, resulting in an isolated zero at the center of the spatial wavevector distribution, indicating energy redistribution into a higher spatial frequency region[27,29].

## Theoretical analysis of isotropic spatial differentiation

We consider a point source positioned at the input plane and transmitting through the focus metasurface. The metasurface has a radius of $30\lambda$ with $\lambda$ representing the acoustic wavelength, and a focal length of $40\lambda$. Figure 2a illustrates the intensity and phase distributions of the acoustic field at the output plane, calculated using the angular spectrum method (refer to Supplementary Note 4 for details)[47]. The intensity pattern reveals an Airy-like spot at the center, whereas the phase profile exhibits an annular distribution. Next, a point source impinging through the 2D spatial differentiator is discussed in Fig. 2b. Here, an observed high-intensity doughnut is accompanied by a spiral phase carrying a 1st-order topological charge. This configuration results in two symmetric points relative to the origin being out of phase but having the same intensity. The point response of imaging systems is characterized by the point spread function (PSF), which is proportional to the Fourier transform of the transfer function $t_{\mathrm{tran}}(x',y')$[36]. Consequently, the PSF of the focus metasurface resembles a diffraction-limited spot, whereas that of the 2D spatial differentiator takes the form of a doughnut with annular intensity and a 1st-order spiral phase. These results confirm the wavevector redistribution brought about by the spiral phase, aligning with the theoretical predictions and resulting in an intensity zero at the center. Figure 2c presents the intensity profiles of the two PSFs along the vertical direction. The peak-to-peak distance of the intensity ring is $1.12\lambda$, and the full width at half maximum of the Airy spot is $0.9\lambda$. Their almost identical values indicate their relation to the imaging resolutions[27]. It is noteworthy that the 2D meta-differentiator exhibits a similar PSF to that of the spiral diffraction gratings[48–53]. When compared with these methods for high-order Bessel beams[48,49], focused vortex beams[50], and perfect vortex beams[51], our meta-differentiator distinguishes itself through its distinctive amplitude and phase modulation characteristics. In addition, this meta-differentiator constitutes a groundbreaking endeavor by integrating orbital angular momentum into the realm of edge-enhanced imaging, an area heretofore unexplored within the field of acoustics, extending beyond conventional applications in communication systems and particle manipulation[54,55]. To illustrate isotropic spatial differentiation, we utilize an amplitude object representing the numeral 2 at the input plane, as shown in Fig. 2d. This object has a height ($h$) of $6\lambda$, a width ($w$) of $3.75\lambda$, and incorporates a slit ($s$) measuring $0.75\lambda$ in width. In the figure, the white region indicates full transmission of acoustic waves, while the black area represents total reflection of acoustic waves. This scenario is relevant for biological tissues or foreign matters that exhibit a significant mismatch in acoustic impedance with the surrounding medium, such as bones or metals[9]. In these cases, the variations in intensity reflect the acoustic contrast. After being processed by the 2D meta-differentiator, the intensity and phase profiles of the acoustic field at the output plane are displayed in Fig. 2e, rotated by 180° about the origin. It is evident that the phase singularity of PSF causes sharp phase discontinuities along the numeral 2 with a $\pi$ difference, resulting in the destructive interference of waves and zero intensity within the numeral 2. Conversely, the phase profile exhibits continuous variations along the edges, leading to constructive interference of waves and allowing for the reconstruction of all edges of the numeral 2. This clearly demonstrates the successful achievement of 2D spatial differentiation, effectively

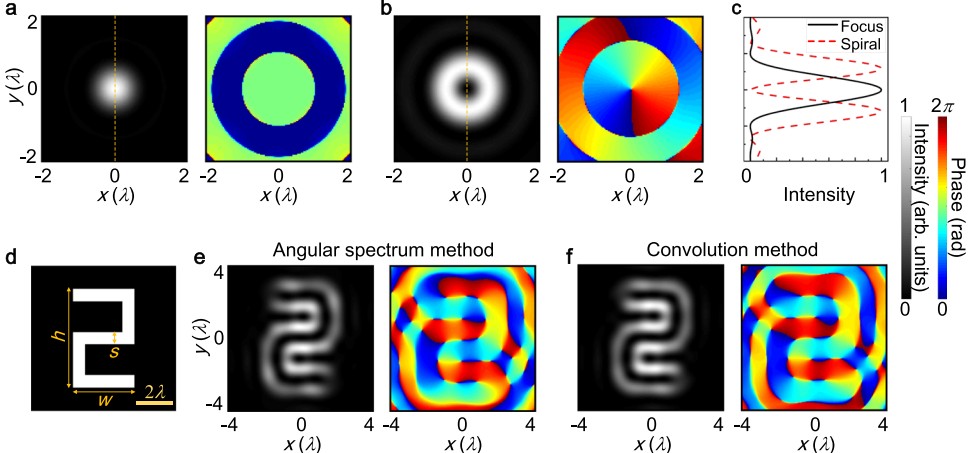

**Fig. 2 | Theoretical analysis of isotropic spatial differentiation.** Acoustic intensity and phase distributions of the PSF for the (**a**) focus metasurface and (**b**) spatial differentiator. **c** Intensity profiles along the vertical direction of the two PSFs. **d** An amplitude object in the shape of the numeral 2 with a height of $6\lambda$, a width of $3.75\lambda$, and a slit of $0.75\lambda$ placed at the input plane. Acoustic intensity and phase profiles at the output plane obtained through the (**e**) angular spectrum method and (**f**) convolution method.

extracting isotropic edge information. Therefore, not exclusively dependent on the intensity profile, the demarcation of object boundaries can be elucidated by discerning the phase profile characterized by a broader width and continuous variations (Supplementary Note 5). Alternatively, the output image can also be obtained by convolving the input image with the PSF of the imaging system, based on Eq. (4). Figure 2f illustrates the calculated profiles of acoustic intensity and phase at the output plane using the convolution method, which closely resemble those in Fig. 2e.

We further investigate the performance of an acoustic metasurface carrying other orbital angular momenta (see Supplementary Note 6). It is found that only the transfer function characterized by ± 1st-order spiral phases is capable of performing 2D differentiation operations on objects for isotropic edge enhancements. Higher-order spiral phases with increased tangential wavevector components are not effective in extracting complete edge information from objects, but could be employed for other specialized purposes[25]. In comparison to 1D edge detection and vertex detection with anisotropic edge enhancements, 2D spatial differentiation excels in presenting the overall outlines of objects[37,45], as detailed in Supplementary Note 7. To validate the universality of the 2D meta-differentiator, we examine the amplitude objects with different numeral and alphabet shapes, see more details in Supplementary Note 8. Additionally, we employ the United States Air Force 1951 resolution test chart to quantify the imaging resolution (see Supplementary Note 9). This confirms that the spatial differentiator excels in isotropic edge detection, achieving a maximum resolution of 1.125 mm at the operating wavelength of 1.5 mm. Supplementary Note 10 also reveals the performance of the 2D meta-differentiator at various excitation frequencies, where the imaging plane dynamically shifts with the variations in excitation frequency, similar to that of the Fresnel zone plates[56,57]. However, our meta-differentiator consistently excels in its core capability of edge detection across a spectrum of frequency adjustments. Furthermore, this 2D meta-differentiator demonstrates the capability to effectively detect objects positioned at different distances, and the resulting enlargement or minification effects can be leveraged to observe minute objects in greater detail (refer to Supplementary Note 11).

### Experimental demonstration of isotropic edge enhancements

We conduct an experiment to verify the 2D isotropic spatial differentiation performed on amplitude objects. To simplify the fabrication process of our experimental sample, we discretize both the amplitude and phase modulators. Detailed discussions regarding the effects of this discretization can be found in Supplementary Note 12. By utilizing 16 amplitude steps together with 2 phase steps, we achieve isotropic edge-enhanced imaging. In our experiments, the operating frequency is 1 MHz, which is commonly used in practical applications of ultrasonic imaging for medical diagnosis and non-destructive testing[16]. The phase-modulated unit, shown in the inset of Fig. 3a, has a thickness ($l$) of 3.75 mm (equivalent to $2.5\lambda$) and a width ($a$) of 1.5 mm (equivalent to $\lambda$). This unit is made of resin material, and stainless steel sheets with a thickness of 200 μm are attached to both lateral sides to reduce unwanted interactions between neighboring units. Figure 3a shows that as the resin height ($h$) increases from 0 to $2.5\lambda$, the phase shift of acoustic waves passing through the phase-modulated unit undergoes a full $2\pi$ change, while the transmission remains relatively high, ranging between 0.8 and 1. In our case, we map 2 phase steps with $\pi$ differences onto the distributions of resin height, specifically 0.45 mm ($0.3\lambda$) and 2.25 mm ($1.5\lambda$), as shown in Fig. 3b. Figure 3c presents the phase meta-grating fabricated with photosensitive resin material by using 3D printing technology with a print resolution of 10 μm (see Methods). The scale bar in the figure represents a length of 22.5 mm. We insert spiral sheets made from stainless steel, with a thickness of 200 μm and a height of 3.75 mm, along the resin parts to provide structural support. On the other hand, the amplitude-modulated unit, as displayed in the inset of Fig. 3d, has a thickness ($d$) of 1.25 mm (equivalent to $0.83\lambda$) and a width ($t$) of 3.0 mm (equivalent to $2\lambda$). This unit is prepared using a steel plate with engraved slits. As the slit width ($w$) increases from 0 to $2\lambda$, the amplitude-modulated unit allows for variations in transmission from 0 to 1, while the phase shift remains almost constant, as shown in Fig. 3d. In this manner, we transform 16 amplitude steps into the distributions of slit width, as illustrated in Fig. 3e. Similar to fabricating the phase meta-grating, we utilize 3D printing technology with a print resolution of 100 μm to fabricate the amplitude mate-grating from stainless steel, as shown in Fig. 3f (see Methods). The cross-sectional views of both the phase and amplitude meta-gratings are presented in Supplementary Note 13. Note that the surface roughness of resin elements in the phase meta-grating measures approximately $R_a = 3.2$ μm, and the amplitude meta-grating has undergone post-processing involving polishing with a surface roughness of approximately $R_a = 6.3$ μm. Both values are significantly smaller than the acoustic wavelength in water (1.5 mm), thereby the impact of roughness on the acoustic field can be safely disregarded. Meanwhile, the sound attenuation for the 0.45 mm and 2.25 mm thick resin elements are about 0.16 dB and 0.80 dB, respectively, while that for the

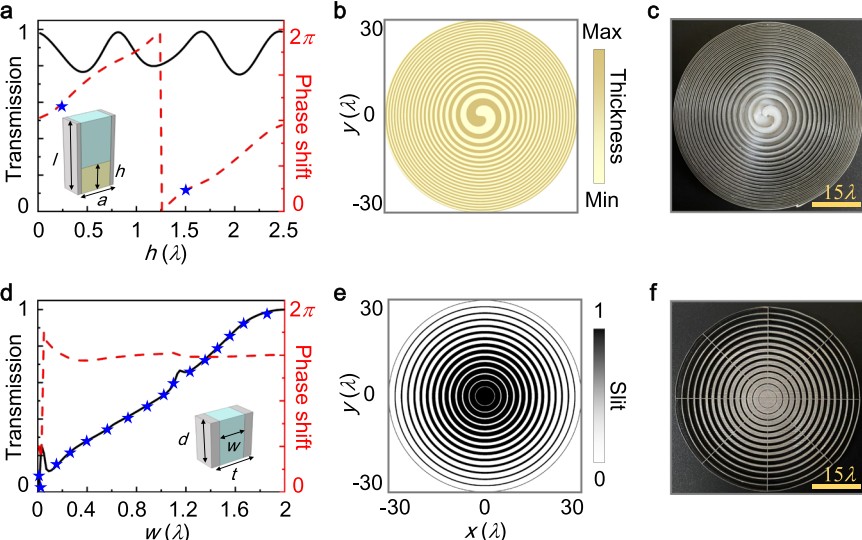

**Fig. 3 | Fabrication of the 2D spatial differentiator. a** Transmission and phase shift of acoustic waves traveling through the phase-modulated unit. Inset: details of the phase-modulated unit. **b** Discrete phase modulations imparted onto the spatial differentiator mapped to the distributions of resin thickness. **c** Photograph of the fabricated phase meta-grating. **d** Transmission and phase shift of acoustic waves traveling through the amplitude-modulated unit. Inset: details of the amplitude-modulated unit. **e** Discrete amplitude modulations mapped to the distributions of slit width. **f** Photograph of the fabricated amplitude meta-grating.

1.25 mm thick steel component is 0.0019 dB, all of which are exceedingly small and can be negligible[58,59].

The experiment setup for isotropic edge-enhanced imaging using the 2D meta-differentiator sample is illustrated in Fig. 4a (see Methods). We present an amplitude object made from a steel plate with a thickness of 1.2 mm and a diameter of 90 mm, as depicted in Fig. 4b. The object features a small circular hole with a radius of 0.5 mm, created using laser etching technology. The ultrasonic waves passing through this hole serve as a point source to confirm the PSF of the spatial differentiator. Figure 4c, d display the simulated and measured acoustic fields, respectively, at the measuring plane of our imaging system. Here, the numerical simulations are carried out by utilizing finite-element-method simulation software COMSOL Multiphysics (see Methods). Both simulated and measured results clearly show an intensity ring along with a clockwise spiral phase. In Fig. 4e, we compare the simulated (line) and measured (circles) intensity profiles along the vertical direction of Figs. 4c and 4d. The measured results agree well with the simulations. Additionally, a hollow pattern is engraved into the steel plate, creating an amplitude object in the shape of the numeral 2, as shown in Fig. 4f. This object has a height of 9 mm ($6\lambda$), a width of 5.625 mm ($3.75\lambda$), and a slit of 1.125 mm ($0.75\lambda$). The corresponding simulated and measured acoustic fields at the measuring plane are presented in Fig. 4g, h, respectively. All boundaries of the numeral 2 are marked by uniform intensity, while arbitrary two points symmetric to the origin are out of phase. Figure 4i showcases the simulated (line) and measured (circles) intensity profiles along the vertical direction of the output images. For quantitative analysis, the contrast is defined as the ratio of the edge intensity to the background intensity[36], and the full width at half maximum (FWHM) of the peak describes the edge width[45]. From the simulations, the contrast of the identified edge to the background is -10−16, and the FWHM is about $0.65\lambda$. The simulated energy conversion efficiency of 2D meta-differentiator, calculated by the proportion of the sound energy at the imaging plane to that at the object plane, is about 0.45. Additionally, the corresponding measured results reveal the contrast and the FWHM are about 10−16 and $0.73\lambda$, respectively, and the energy efficiency is estimated to be 0.27. The measurements and simulations exhibit good agreement, although slight deviations may be attributed to fabrication errors in the experimental sample. Despite the energy efficiency is

subject to some reductions due to the reflections of meta-differentiator, the absolute intensity of the resultant image at the output plane can be effectively enhanced by augmenting the power of incident waves. The edge enhancements of amplitude objects in different sizes processed by the 2D spatial differentiator are also experimentally demonstrated, as detailed in Supplementary Note 14. Additionally, the experimental results of numeral 3 confirm that the 2D meta-differentiator performs well on amplitude objects in different shapes (refer to Supplementary Note 15). Furthermore, our meta-differentiator serving as an image processor can incorporate principles applicable to both transmission- and reflective-modes, enabling augmented observational capacity through the comprehensive extraction of edge information from objects (see Supplementary Note 16). Through a strategic rearrangement of incident waves and detected objects, the transmission-mode meta-differentiator also can effectively process backscattering waves from objects, extracting all object edges, as elaborated in Supplementary Note 17.

## Edge detection of phase objects

We conduct further investigation into edge-enhanced imaging of phase objects, which is often the case for biological materials with acoustic properties similar to that of ambient medium, such as cells or soft tissues[14]. The acoustic waves passing through phase objects have almost uniform intensity, while the shifted phases due to the stiffness of objects lead to acoustic contrast[14]. Edge contrast imaging is potentially significant for discerning such objects that have slight phase contrast with the ambient medium[12]. Figure 5a illustrates an ideal phase object in the shape of the numeral 2. It has a height of $6\lambda$, a width of $3.75\lambda$, and a slit of $0.75\lambda$. The incident Gaussian plane wave has a width of 12.7 mm and undergoes a phase advance of a $2\pi/3$. This phase advance is employed to generate a phase difference of $2\pi/3$ between the phase object and the surrounding medium. Figure 5a illustrates the simulated acoustic intensity and phase distributions at the output plane. It is evident that the contour lines between different phases along the numeral 2 are enhanced, regardless of their orientations. This indicates the remarkable ability of the spatial differentiator in isotropic phase contrast imaging. The circular edges around the numeral 2 correspond to the edge enhancements caused by the amplitude jump of the incident Gaussian plane wave. In Fig. 5b, we

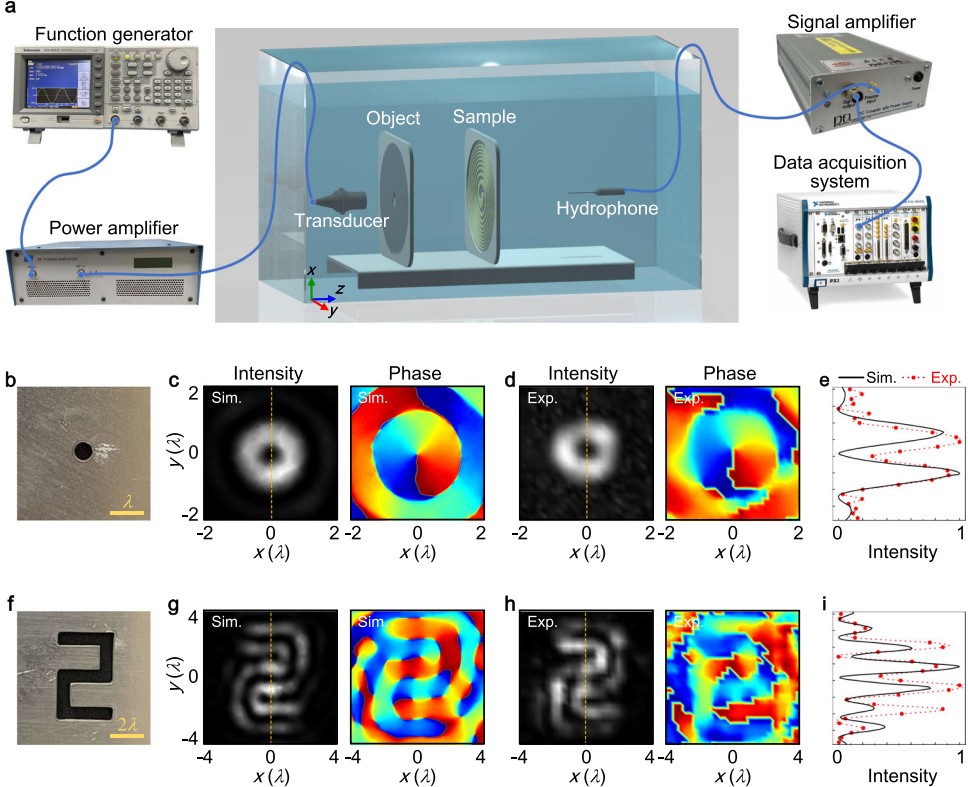

**Fig. 4 | Experimental demonstration of isotropic edge enhancement.**
**a** Schematic illustration of the experimental setup for underwater acoustic field measurements. **b** Amplitude object fabricated from a steel plate with a circular hole. **c** Simulated and (**d**) measured acoustic intensity and phase distributions at the output plane under the illumination of a point source from the input side.

**e** Simulated (line) and measured (circle) intensity profiles along the vertical direction of the output images in (**c**) and (**d**). **f**–**i** Same as (**b**–**e**), but for the amplitude object made of a steel plate with a hollow pattern in the shape of the numeral 2 at the center.

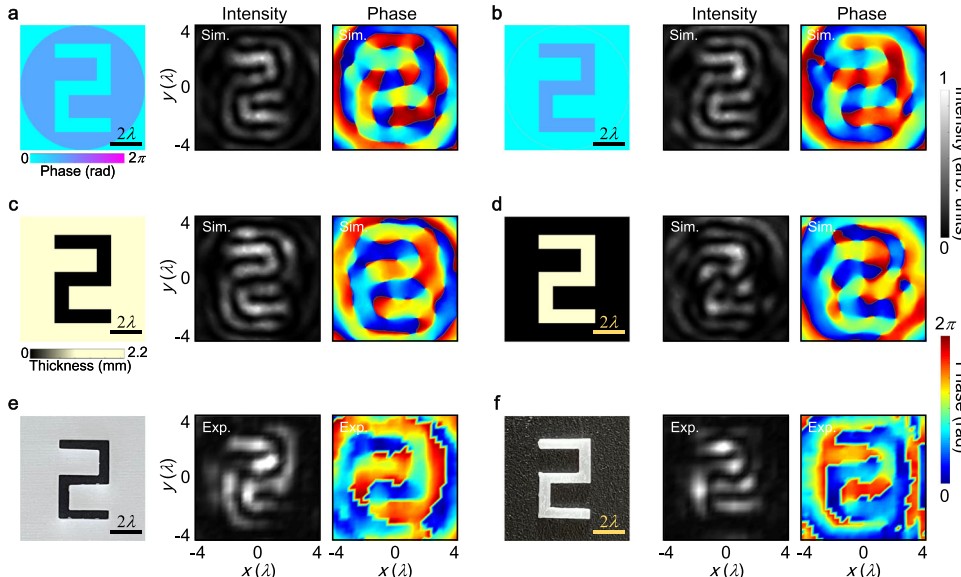

**Fig. 5 | Edge detection of phase objects.** Ideal phase object in the shape of the numeral 2 with (**a**) a phase delay of $2\pi/3$ and (**b**) a phase advance of $2\pi/3$ in relation to the ambient medium. Middle and right panels: simulated acoustic intensity and phase distributions at the output plane. **c, d** Similar to (**a, b**), but for the full-wave simulated phase objects made of resin material with a thickness of 1.2 mm. **c** shows a hollow pattern of the numeral 2 at the center, while panel (**d**) shows the complementary physical numeral 2. **e, f** Same as (**c, d**), but for the fabricated phase object samples made of the same material and parameters as in (**c, d**).

demonstrate the accurate edge detection of a phase object with a phase advance of $2\pi/3$ in relation to the background medium. Supplementary Note 18 also includes a discussion on the phase objects with varying phase gradients compared to the ambient medium. Note that the edge detection of phase structures can be achieved through the application of digital image processing techniques on the acquired image obtained from the broadband ultrasonic pulse time-of-flight (TOF) measurements[1,14]. However, our meta-differentiator, grounded in the principles of computational metamaterials, exhibits unique advantages encompassing low power consumption, high-speed processing capabilities, and real-time imaging proficiency, rendering it particularly valuable in contexts requiring dynamic object monitoring[33,34,41]. In addition, the lock-in analysis employing narrowband signals excels in detecting phase structures as well, offering heightened sensitivity in phase measurement and serving as a robust methodology for extracting subtle signals from noise[60]. Despite of its advantages in scenarios where both high sensitivity and selectivity are imperative, this method is accompanied by considerations of complexity, cost, and the prerequisite for stable reference signals. In contrast, our spatial differentiator serving as an image processing tool shows the specific capability of directly extracting edge information from objects to facilitate recognition, rather than focusing on noise mitigation for signal enhancement.

To prepare the practical phase objects, we fabricate a circular plate using resin material, measuring 90 mm in diameter. This plate features a hollow pattern of the numeral 2 at its center, as depicted in Fig. 5c. The thickness of the resin plate is 1.2 mm, which allows for full transmission of ultrasonic waves but introduces a phase delay of $2\pi/3$ in the hollow area. In Fig. 5c, the simulated acoustic fields at the output plane of the imaging system showcase isotropically enhanced intensity distribution along all edges of the numeral 2. For further investigation, we examine a complementary physical numeral 2 with the same thickness of 1.2 mm, for achieving a phase advance of $2\pi/3$ relative to the ambient medium (Fig. 5d). The interior edges of the numeral 2 exhibit significant enhancement, while the outer edges appear slightly ambiguous due to interactions with incident acoustic waves. This 2D meta-differentiator allows for easy identification of the contours of phase objects, facilitating the discrimination of objects with phase differences from the background environment. In Fig. 5e, a phase object sample is fabricated using 3D printing technology, with the same materials and parameters as in Fig. 5c. The corresponding acoustic intensity and phase profiles, measured at the output plane, are consistent with the simulations. To validate the simulated results (Fig. 5d), we fabricate a phase object in the shape of the numeral 2, as depicted in Fig. 5f. The measured acoustic intensity distribution agrees well with the simulation. Note that in the imaging system without the employment of spatial differentiator, discerning amplitude and phase objects from the surrounding medium proves challenging, owing to the unavoidable sound diffractions in the far field (refer to Supplementary Note 19 for detailed control experiments). The incorporation of acoustic meta-differentiator, however, enables the reconstruction of far-field images for both amplitude and phase objects obviating the need for post-processing reconstruction algorithms, characterized with enhanced edge features that facilitate straightforward identification. Additionally, we numerically explore the phase objects in different shapes, including numerals and alphabets, with a $\pi$ phase shift relative to the ambient medium (see Supplementary Note 20). The spatial differentiator is proved to be capable of revealing more detailed boundary information of phase objects, as experimentally demonstrated with the results of a phase object in the shape of the numeral 3 in Supplementary Note 21. Compared to the conventional imaging systems based on focus metasurfaces, this 2D spatial differentiator exhibits good performance in presenting more detailed information of the phase objects with slight phase variations to the surrounding medium (Supplementary Note 22). Importantly, our spatial differentiator is not merely an edge extractor, but an image intensifier that enhances contrast of the phase objects that are hardly distinguishable from the ambient medium in conventional imaging systems.

## Application potential in biological imaging

The contrast of ultrasound imaging is limited, making it challenging to achieve clear visualization of small structures like fetal fingers or auricles in early stage of pregnancy. There is a strong desire for advanced image processing techniques to enhance the edges and contrast of fingers or auricles from amniotic fluid for more accurate medical diagnosis. The edge detection method, sensitive to slight variations in acoustic properties, has the potential for high-contrast ultrasonic imaging of biological samples. For illustration, Fig. 6a presents an artificial mini hand made of resin material with a thickness of 1.2 mm, featuring a stretched out forefinger. The scale bar represents a length of 3 mm. The simulated intensity profile demonstrates that the outlines of the mini hand are enhanced in all orientations, allowing for the extraction of detailed edge information from the forefinger. Figure 6b shows a photograph of the fabricated mini hand using the same materials and parameters as in Fig. 6a. The measured result reveals that all edges of the mini hand are effectively emphasized. Furthermore, in Fig. 6c, we compare the simulated (line) and measured (square) intensity profiles of the acoustic fields in Fig. 6a, b, respectively, and find a close match between them. We also consider an artificial mini hand with both the forefinger and midfinger stretched out, as shown in Figs. 6d–f. The sketches of the mini hand highlight the edge information of both fingers, which can be easily obtained. We also experimentally investigate the edge detection of 3D artificial mini hands in Supplementary Note 23. It is observed that the outer boundaries of 3D mini hands are clearly delineated, while the non-comparable isotropic characteristics within the intensity profiles are attributed to depth variations between the fingers. We also discuss the application of this meta-differentiator in the context of imaging 3D structural objects, as detailed in Supplementary Note 24. It reveals that our method is well-suited for the edge detection of 3D objects featuring sparse interior slice structures; however, it may encounter difficulties when applied to processing 3D objects characterized by densely packed slice structures. To further enhance the capabilities of 3D structural imaging, our acoustic meta-differentiator is expected to be integrated with either the modern ultrasonic computed tomography (CT) technique or the TOF method[1,14]. In the ultrasonic CT technique, the meta-differentiator together with an ultrasonic transducer is used to obtain a series of cross-sectional slices with edge enhancements from diverse perspectives, which are subsequently processed via image post-processing algorithms to reconstruct 3D objects with edge-enhanced contours. In the TOF method, the meta-differentiator is envisaged to process the resulting echoes induced from interactions of ultrasound pulses with tissue structures at varying depths. Based on the amplitude information and time intervals between the emission of ultrasonic pulses and the reception of each echoes, it becomes feasible to reconstruct the edge contours of 3D objects.

Additionally, we discuss the performance of this spatial differentiator on other artificial biological models. Figure 6g presents a silicone urethral catheter with an inner diameter of 2 mm and an outer diameter of 3 mm. The corresponding experimental result shows two lateral edges of the urethral catheter, which are 2.5 mm apart. We then insert a small metallic ball with a diameter of 3 mm inside the urethral catheter to simulate a foreign object in the human urethra. As shown in Fig. 6h, the measured intensity profile displays not only all edges of the urethral catheter but also the bulging part caused by the metallic ball. The intensity profiles in Fig. 6i reveal the presence of a foreign object with a diameter of 3 mm inside the urethral catheter. Moreover, we prepare an artificial blood vessel model with a diameter of 5 mm as an object placed at the input plane, as shown in Fig. 6j. Due to the ultra-

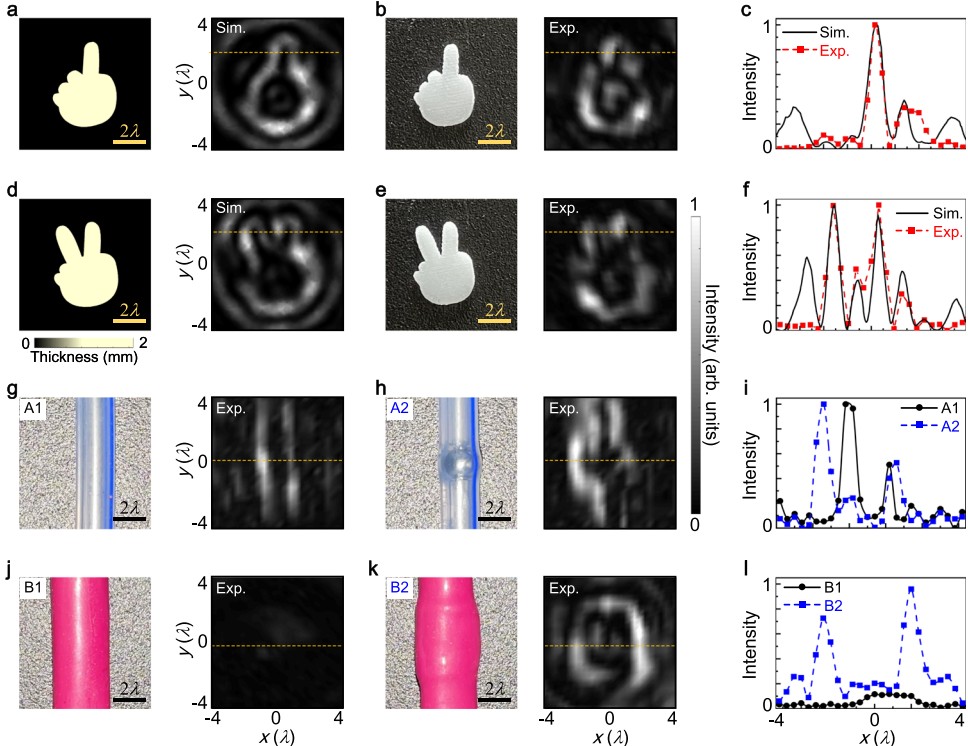

**Fig. 6 | Edge detection of artificial biological models. a** An artificial mini hand with the forefinger stretched out, along with the simulated acoustic intensity profile at the output plane. **b** Photograph of the mini hand with the same material and parameters as in (**a**), and the measured intensity profile at the output plane. **c** Intensity profiles along the horizontal direction of the simulated and measured results in (**a**) and (**b**). **d–f** Same as (**a–c**), but for the artificial mini hand with both the forefinger and midfinger stretched out. **g** Photograph of a silicone urethral catheter, and the corresponding measured intensity profile at the output plane.

**h** Photograph of the silicone urethral catheter with a small metallic ball inserted inside, and the measured intensity profile at the output plane. **i** Intensity profiles along the horizontal direction of the measured results in (**g**) and (**h**). **j** Photograph of an artificial blood vessel, and the measured intensity profile at the output plane. **k** Photograph of the artificial blood vessel with a large blood clot inside, and the measured intensity profile at the output plane. **l** Intensity profiles along the horizontal direction of the measured results in (**j**) and (**k**).

thin wall thickness of the blood vessel and its similar acoustic properties to the surrounding medium, the edges of the blood vessel are nearly invisible in the measured result, making it challenging to differentiate them and resulting in low contrast in the images. In Fig. 6k, we roll up a small piece of the blood vessel and placed it inside another blood vessel, simulating a blood clot in a blood vessel. The measured intensity profile clearly shows the outer boundaries of the blood clot. Additionally, the intensity line in Fig. 6l indicates that the approximate diameter of the blood clot is about 5.8 mm. The general shape and boundary of the artificial biological models are remarkably discernible with high contrast. It is noteworthy that several limitations in the measurement configuration may impede the current applications of spatial differentiator. Primarily, the field-scanning method employing one single hydrophone together with translation stage is time-consuming. As an alternative, the Schlieren imaging method is promising for real-time visualization of acoustic fields[61], potentially capturing the dynamic motions of living organisms[7]. Additionally, our current configuration faces challenges in imaging 3D objects, and necessitates the imperative integration of contemporary imaging techniques to facilitate comprehensive 3D object measurement[14]. Furthermore, the reflective-mode imaging system combined with ultrasonic pulse method should be constructed for commoner applications in medical diagnosis[13]. Hence, there is an opportunity for further refinement of our methodology to enhance its applicability in practical biomedicine and engineering scenarios.

## Discussion

In conclusion, we have developed an underwater 2D isotropic edge-enhanced ultrasonic imaging technique by utilizing an acoustic meta-differentiator. This technique improves the image contrast of objects without the need for contrast agents or external physical fields. The spatial differentiator is comprised of an amplitude modulator with linear transmission in the radial direction, and a phase modulator that combines both focus phase and spiral phase with a topological charge of 1. By applying the generalized Snell's law, the spatial differentiator redistributes the wave vector to higher values, leading to a central zero in the wavevector space for 2D isotropic differentiation operations. The PSF of the meta-differentiator exhibits an intensity ring with a spiral phase. As verified by theoretical analysis, the output image can be acquired by convolving the input object image with the PSF of the meta-differentiator. Both numerical simulations and experimental results confirm that the spatial differentiator enhances the edges of amplitude objects in an isotropic manner. Furthermore, we demonstrate successful edge detection of phase objects and artificial biological models, revealing that the meta-differentiator excels in visualizing the objects with slight variations to the ambient medium that are indistinguishable by conventional imaging systems.

Importantly, our 2D meta-differentiator offers distinct advantages over the existing computational acoustic metamaterials in several key aspects (Supplementary Note 25). Primarily, this meta-differentiator with inherent 2D spatial differentiation operations can realize the simultaneous extraction of all object boundaries at the imaging plane directly, without post-processing procedures like point-to-point scanning and spatial spectrum superposition[45,46]. Meanwhile, this design leverages a single planar metastructure, incorporated into both transmission- and reflective-mode imaging systems in free space, significantly reducing the overall system dimensions and enhancing compatibility with compact integrated systems[42,43]. Furthermore, our

work provides an underwater imaging methodology for efficiently extracting edge information from both amplitude and phase objects, which holds substantial promise for the advancements in biomedicine and engineering applications.

Building upon the advancements in acoustic microscope and ultrasound CT technology, our meta-differentiator introduces an edge-enhanced imaging technique as a valuable complement to conventional bright-field imaging. This innovative approach, which improves the visibility of object boundaries and accentuates regional distinctions within images, can find applications in biomedical imaging and nondestructive testing. In the context of medical imaging, edge enhancement holds significant potential for enhancing the visualization of anatomical structures and abnormalities, thereby contributing to the early detection of diseases, facilitating surgical planning, and augmenting the interpretation of medical imagery. In industrial applications, edge-enhanced imaging can serve as an indispensable tool for identifying defects, cracks, and irregularities in manufactured products, thereby enhancing quality control and inspection procedures. Conclusively, our meta-differentiator is promising to be seamlessly integrated with acoustic microscopes or ultrasound CT techniques to deliver superior object recognition capabilities.

## Methods

### Numerical simulation

The numerical simulations were conducted using finite-element-method simulation software COMSOL Multiphysics. In the simulation domain, the input plane, acoustic metasurface, and the output plane were aligned and located at positions with an interval of $f = 40\lambda$ along the propagation direction. The radius of the acoustic metasurface was $30\lambda$ at the operating frequency of 1 MHz. The phase meta-grating in the spatial differentiator exhibited a spiral distribution with the mathematical expression of $r(\theta) = \sqrt{\left(f + \frac{\theta}{2\pi}\frac{\lambda}{2}\right)^2 - f^2}$. It was made of photosensitive resin material with a mass density of $\rho_r = 1176\ kg/m^3$ and a longitudinal sound speed of $c_r = 2460\ m/s$. Stainless steel sheet with a mass density of $\rho_s = 7800\ kg/m^3$, longitudinal sound velocity $c_{sl} = 6100\ m/s$, and transverse sound velocity $c_{st} = 3300\ m/s$ was attached to both lateral sides to reduce interactions between neighboring phase-modulated units. The amplitude meta-grating was prepared from a stainless steel plate with a radius of $30\lambda$, which was engraved with 14 annular slits of gradually varied widths along the radial direction. The ambient medium was water with a mass density of $\rho_w = 1000\ kg/m^3$ and a longitudinal sound speed of $c_w = 1500\ m/s$. When an amplitude object was placed at the input boundary of the simulation domain, the imaging region was set as plane wave radiation condition with an amplitude of 1 Pa, while the remaining regions were modeled as acoustic rigid boundaries to prevent sound diffractions. However, for a phase object, the imaging area and the input Gaussian plane waves were set as plane wave radiations with an amplitude of 1 Pa but with different initial phases. To eliminate the influences of sound reflections, a plane wave radiation condition was imposed on the output boundary, and cylindrical wave radiation conditions were set on the lateral boundaries of the simulation domain.

### Sample fabrication

The fabricated phase meta-grating was composed of two components crafted from photosensitive resin and stainless steel. Due to current technological constraints, the fabrication process of the phase meta-grating was divided into two separate steps. Initially, the resin-based component was manufactured using stereolithography-based 3D printing technology (microArchR S240, projection micro stereolithography; resolution 10 μm; BMF Precision Tech Inc., https://www.bmftec.cn/3d-printers/s240). Stereolithography employed photosensitive resin material that can be solidified through exposure to UV laser. Then, a precisely controlled UV laser directed its beams into the resin reservoir along a predetermined path to polymerize the photocurable resin into a 2D patterned layer. After each layer solidified, the platform descended and another layer of uncured resin was prepared for patterning. Upon completion of the printing process, all printed components undergone post-curing to attain maximum strength and stability. Additionally, rough surfaces with unevenness were meticulously sanded to achieve a smooth finish with the roughness of about $R_a = 3.2$ μm. Next, the steel-based component, made of stainless steel sheet with a thickness of 200 μm, was strategically inserted into the interspace between neighboring resin units to minimize undesired sound interactions.

The amplitude meta-grating made of stainless steel was fabricated through 3D metal printing technology, specifically powder bed fusion (EOS M 100, direct metal laser sintering; resolution 100 μm; GuangZhouXinYuan Metals&TCO., http://xy-metal.com). This additive manufacturing method used a precisely controlled laser beam to scan the steel powders along a predetermined path and sinter them through localized heating. Under the influence of high-powered lasers, neighboring powder particles fused together via molecular diffusion, enabling the subsequent layering process to commence. Upon the completion of amplitude meta-grating, post-processing treatments were taken such as cutting, machining, hot isostatic pressing, or solution annealing, all of which served to enhance the overall quality of the fabricated components. As a finishing step, the roughness of amplitude meta-grating was refined to about $R_a = 6.3$ μm through the grinding and polishing procedures.

### Experimental measurement

We conducted measurements of the acoustic field in an anechoic water tank with dimensions of $1000 \times 740 \times 600\ mm^3$. An ultrasonic transducer (Olympus V303-SU) with a diameter of 0.5 inches (12.7 mm) was connected to a multi-function waveform generator (Tektronix AFG3052C) to convert electrical signals into ultrasonic waves. To ensure uniform acoustic fields, we placed an imaging object at a distance of 3 mm away from the transducer. The fabricated meta-differentiator sample, composed of amplitude and phase meta-gratings, was aligned and positioned 60 mm away from the object. To perform ultrasound field scanning at the measuring plane, located at a distance of 60 mm from the sample, we employed a needle hydrophone with a diameter of 0.5 mm connected to a translation stage. This Hydrophone (PA NH0500, output impedance 18 pF ± 3 pF; typical probe sensitivity −250.5 dB re 1V/μPa; frequency range 0.1–20 MHz; Precision Acoustics Ltd, UK) was coupled with submersible preamplifier and DC coupler to configure an integrated hydrophone system exhibiting an output impedance of 50 Ω. For more comprehensive insights into the specifications of the needle hydrophone, further details can be referenced in (https://www.acoustics.co.uk/product/0-5mm-needle-hydrophone/). The other hydrophone identical to the first one was positioned in the vicinity of the transducer to serve as the phase reference. The data acquired from two hydrophones were amplified, then recorded using a digital storage oscilloscope (NI PXI-5152), and subsequently processed using Matlab 2013 software. Since the water used in experiments was purified and degassed, we could disregard the effects of impurities and air bubbles on the ultrasound fields.

## Data availability

All technical details for producing the figures are enclosed in the Supplementary Information. Data are available from the corresponding authors H.L., Y.C. or X.L. upon request.

## Code availability

All technical details for implementing the simulation are enclosed in the Supplementary Information. Codes are available from the corresponding authors H.L., Y.C. or X.L. upon request.

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

## Acknowledgements

This work was supported by the National Key R&D Program of China (2022YFA1404400), NSFC (12225408, 12227809, 12074183, 12134002, and 12004176) and the Fundamental Research Funds for the Central Universities (020414380181). Y.J. acknowledges the funding support from the Postgraduate Research & Practice Innovation Program of Jiangsu Province (KYCX23-0095), and the scholarship support from the China Scholarship Council. The authors thank Dr. Xin Liu (Fudan University) for fruitful discussions.

## Author contributions

Y.J. and Y.C. initiated the project and conceived the idea. H.L., Y.C., and X.L. guided the research. Y.J. carried out the theoretical analyses and conducted finite-element-method simulations. Y.J., H.L., and Y.C. designed the experimental scheme. Y.J., S.Z, X.Z., and H.L. conducted the measurements. Y.J., S.Z, X.Z., H.L, and C.X. assisted with sample fabrication. C.X. and M.D. measured the mechanical property of the sample. C.X., Y.B., D.W., and M.D. provided insight and interpretation of the edge enhancement. Y.J., H.L., Y.C., C.W.Q., and X.L. wrote the manuscript. All the authors contributed to the discussions of the results and the manuscript preparation.

## Competing interests

The authors declare no competing interests.
