## [Peer Review File · Nature Communications]

Compact meta-differentiator for achieving isotropically high-contrast ultrasonic imagingReviewer #1 (Remarks to the Author):

This manuscript described an acoustic meta-differentiator based on vortex pattern to enhance intensity contrast of transmitted waves for ultrasound imaging. The physical principle as claimed by the authors was due to the added wavevector of the vortex pattern from the meta-differentiator that scattered the image information to a high k component. The claims of the detection of images by transmitted waves through masks and meta-differentiator were supported by numerical and experimental studies. The manuscript is well-written, and the results are interesting. However, there are three major concerns to be addressed.

1. A major concern I have is the lack of control experiments to provide direct evidence of the improved edge detection as claimed by the authors. All the simulations and experiments were through both the target masks and the meta-differentiator. There are no control experiments on the imaging of the target masks without transmitting through the meta-differentiator. Thus, it is difficult to know how much edge contrast enhancement the meta-differentiator achieved. Therefore, the significance of the meta-differentiator is vague.

2. Another major concern is about the potential of the technology in practical applications of ultrasound imaging. In the clinical setting, the ultrasound machines reconstruct images based on backscattering of waves from impedance changing surfaces. This is true for A mode, B mode, and M mode imaging. The meta-differentiator proposed by the authors is based on the transmitted waves, which is usually used in bright field imaging of optical microscopy and X-ray imaging. Thus, it is hard to image how the proposed technique can be applied for practical ultrasound imaging, which dramatically reduces the significance of the work.

3. There is also a concern about the physical principle of the meta-differentiator. While sharp edges lead to high k components in the scattered wave, the proposed technique of further increasing the wavevector magnitude has a risk of scattering imaging information to near-field components, and thus affecting the resolution or distort the image. This needs to be discussed.

Besides the three major concerns, there are several other minor issues to be addressed, like the "biomedical and engineering" in the first sentence of the abstract should be biomedical engineering.

Reviewer #2 (Remarks to the Author):

The authors proposed using acoustic metasurfaces to enhance isotropic edge boundary imaging in immersion. The method employs amplitude and phase modulation to highlight object features. This work can prove to be a gateway for the practical applications of using it to combat contrast issue. However, since the approach utilizing amplitude and phase modulation has already been theoretically explored in existing studies, this research is not entirely novel. Additionally, the absence of comparative analysis with various approaches to edge boundary detection makes it challenging to consider this paper for publication. Furthermore, one of the primary reasons is the complete absence of proposals regarding potential applicability to existing ultrasound imaging equipment. Nevertheless, considering that the paper effectively substantiates its theoretical foundation with experimental support and holds several implications of significance, I highly recommend transferring this work to another journal.

Some of the following details also should be addressed to be published in the other journal.

Comment #1 : A comprehensive account of the production of the suggested metasurface for experimental validation is absent. It is imperative to elucidate the specifics concerning the material, manufacturing technique, and production process to a degree that allows readers to replicate it

accurately.

Comment #2 : In this paper, the theoretical exposition of enacting the suggested concept appears rather robust. However, the methodology proposed herein does not introduce novel theories but rather elucidates pre-existing ones, thus leaving the extent of advancements in comparison to established literature somewhat unclear. Moreover, though experimental implementation and validation were undertaken, the account of experimental intricacies remains notably inadequate. Especially concerning metamaterials, as the practical realization holds more significance and complexity than theoretical advancement, a more explicit and detailed elaboration of the metasurface is imperative to effectively manifest the research team's claimed theory.

Comment #3 : While the paper demonstrates the feasibility of realizing high-contrast ultrasonic imaging through the metasurface proposed, the results section solely offers a qualitative comparison and evaluation of simulations and experiments. A quantitative assessment of the heightened contrast and resolution achieved by the proposed approach would be pertinent in elucidating and accentuating the authors' proposition. Furthermore, to accentuate the enhancement and merit of the concept posited in this study, it is deemed essential to undertake quantitative comparisons with prior research or the state-of-the-art meta-differentiators.

Comment #4 : For Figure 3b on page 21, you have presented the discrete phase modulations applied to the spatial differentiator, correlated with the distributions of resin thickness. It would be advisable to include a cross-sectional view or data on surface roughness to assess and authenticate your manufactured metasurface.

Comment #5 : Figure S14 (b & c) depicts an isotropic representation of the finger boundary, whereas Figure S14 (d & e) does not manifest comparable isotropic attributes within the intensity profile. This variance could potentially be ascribed to variations in depth between the fingers. Considering that a considerable portion of the showcased measurements and simulations are based on planar structures, an important query emerges: Does the Meta-differentiator face obstacles when employed for three-dimensional structural imaging?

Comment #6 : To ensure the wider applicability of the Meta-differentiator in biological imaging, incorporating three-dimensional structural imaging becomes crucial. Could the authors provide further insight into this facet and deliberate on prospective remedies or avenues for additional research aimed at resolving this concern?

Comment #7 : There seems to be a lack of comprehensive elaboration regarding the theoretical importance of phase in comparison to amplitude for edge detection, along with its practical manifestation. In terms of quantitative analysis, is there a discernible enhancement when employing phase advancements and phase delays for material determination, as opposed to relying solely on intensities? Further details are required concerning the interpretation of phase information in both simulations and experiments, particularly addressing how the images deviate from the anticipated outcomes.

Comment #8 : To the right-hand side of Fig. 2a, there exist four crimson extensions stemming from the central circle at 45, 135, 215, and 305 degrees. What factors contribute to their presence (e.g., simulation boundary conditions and artefacts) and what is their contextual significance? Furthermore, there is an additional red contour on the outer edge of the outer circle. What does this phase delay indicate?

Comment #9 : The acoustic metasurface is constructed using resin and a stainless steel sheet. It would be necessary to measure attenuation when ultrasound signals pass through them, and specific details about the absolute magnitude of the ultrasound pressure signals administered would be essential. While most intensity values have been normalised solely to arbitrary units, comparing the differences in specific absolute pressure values could offer crucial insights for future applications.

Comment #10 : Ascertaining the results necessitates precise details about the specifications of the receiving hydrophone. Providing more specific information regarding the hydrophone employed is crucial. The model number alone might not be adequate for locating the specifications online.

Reviewer #3 (Remarks to the Author):

Key results:

The manuscript presents and demonstrates a method to realize a "meta-differentiator for achieving isotropically high-contrast ultrasonic imaging".

The authors describe how to realize a prescribed spatial operation by modulating the amplitude and phase of an ultrasonic beam passing through a circular plate and show how a phase modulator composed of focus and spiral phases with a first-order topological charge can act as edge detector in ultrasonic imaging.

In this regard, the results are innovative to my knowledge.

However, I have two main comments about the proposed "metasurfaces" and the overall novelty of the results that the authors should address:

1) Structures very similar to the one described in the article are known in literature as "spiral diffraction gratings" or "spiral Fresnel zone plate", depending on the characteristic of both amplitude and phase modulation, and have been used for generating or manipulating high-order acoustic or optical Bessel beams or to generate acoustic vortexes.

Not being exhaustive, see for instance:

a. Jiménez, N., Picó, R., Sánchez-Morcillo, V., Romero-García, V., García-Raffi, L. M., & Staliunas, K. (2016). Formation of high-order acoustic Bessel beams by spiral diffraction gratings. *Physical Review E*, 94(5), 053004

b. Jiménez, N., Romero-García, V., García-Raffi, L. M., Camarena, F., & Staliunas, K. (2018). Sharp acoustic vortex focusing by Fresnel-spiral zone plates. *Applied Physics Letters*, 112(20).

c. Jiménez-Gambín, S., Jimenez, N., Benlloch, J. M., & Camarena, F. (2019). Generating Bessel beams with broad depth-of-field by using phase-only acoustic holograms. *Scientific reports*, 9(1), 20104.

d. Vaity, P., & Rusch, L. (2015). Perfect vortex beam: Fourier transformation of a Bessel beam. *Optics letters*, 40(4), 597-600.

e. Zhang, B., & Zhao, D. (2010). Focusing properties of Fresnel zone plates with spiral phase. *Optics Express*, 18(12), 12818-12823.

The devices presented in the references mentioned above implement PSFs very close to that mapping the spot in Figure 2a in that of Figure 2b of the manuscript, i.e. the omni-directional differentiator.

For instance in refs. (a) and (b) various spiral diffraction gratings have been developed and tested to generate non-zero orbital angular momentum beam starting from standard piston-transducers (acoustic) or Gaussian beams (optics), i.e. approximating the transformation of an ideal Bessel J0 beam into a J1 beam

Consider also that the PSFs reported in supplementary material note 6 are actually the patterns of Hermite-Gauss beams, which can be generated from a Gaussian beam with spatial derivative in x- or y- direction and are very similar to the pattern generated with spiral diffraction gratings in [Yu, X., Trallero-Herrero, C. A., & Lei, S. (2016). Materials processing with superposed Bessel beams. *Applied Surface Science*, 360, 833-839.] as a superposition of different Bessel beams.

The authors should therefore include these topics/works in the review of the state of the art and elaborate on and highlight the differences and the novelty of their structure and method. Linking the manuscript results with the theory of Bessel beams can also add value to the paper, as there are many applications related to the manipulation of acoustic beams with orbital angular momentum.

2) According to [Assouar, B., Liang, B., Wu, Y., Li, Y., Cheng, J. C., & Jing, Y. (2018). Acoustic metasurfaces. *Nature Reviews Materials*, 3(12), 460-472.], it is generally agreed that "Acoustic metasurfaces are 2D materials of subwavelength thickness capable of providing non-trivial local phase shifts (or amplitude modulation) or extraordinary sound absorption. The uniqueness of metasurfaces lies in their ability to freely tailor the wave fields such that the phase and/or amplitude is fully controlled. The acoustic metasurface concept is based on arrays of subwavelength units, including (but not limited to) Helmholtz resonators, membranes and coiling-up space structures..".

The present structure does not exploit any resonance phenomenon, nor does it achieve sub-

wavelength thickness (thickness equivalent to 2.5λ), so the term metasurface is in my opinion not adequate, while instead spiral diffraction grating or just diffraction grating seems more fitted, especially in the light of Eq.4 and its interpretation as a convolution with the Fourier transform of t_{tran} . Diffraction gratings indeed manipulate vectors in the k-space as the present structure does. To clarify the distinction between metasurfaces and diffraction gratings, and to provide a further reference closely linked to the manuscript results, see for instance [Jiang, X., Li, Y., Liang, B., Cheng, J. C., & Zhang, L. (2016). Convert acoustic resonances to orbital angular momentum. Physical review letters, 117(3), 034301.] where a 0.5λ metasurface is used to apply a spiral phase shift with a 1st order topological charge.

In this paper the metasurface applies a spiral phase shift by combining different resonating structures transforming a gaussian beam in one having orbital angular momentum.

Validity:

The analytical, numerical and experimental analysis of the phase modulator performance is very accurate and clear, and the results reported both in the manuscript and in supplementary material very good

Significance:

The use of the present structure in ultrasonic diagnostic or NDT imaging is not straightforward. As the authors considered just a single frequency excitation, it is not clear if the structure can work with different frequencies or, as for the Fresnel zone plates, the proposed structure works mostly at single frequency and fixed distance, i.e the focus moves by changing the excitation frequency.

If this is the case, the use of the present structure in ultrasonic imaging for diagnostic or NDT purposes is not feasible in most of the applications, which usually requires broad-band signals and a single-side inspection.

Instead, the structure could be successfully used in through transmission imaging with narrowband excitation.

This fundamental aspect should be discussed in the paper.

Further, if the structure can instead work with a broadband signal maintaining the focal distance, this should be emphasized as this is one of the main goals in ultrasonic imaging and experimental results obtained by using a broadband signal should be added.

In respect to the edge detection of "phase-structures", it is not true that they could be so hard to detect. By using a short pulse excitation with a broadband spectrum, such a structure can be easily detected by measuring the Time-of-flight of the ultrasonic pulse passing through the structure and then edge detection processing can be applied to the resulting image. By using narrowband signals, the phase structure can be detected as well, for instance by using the lock-in analysis that can provide a very high sensitivity in phase measurement.

However, the direct imaging of the edges of phase-structure is undoubtedly a very good result.

Data and methodology:

The methodology followed by the authors is sound, the quality of the data and presentation is very good as well.

Suggested improvements

As said before, there is a lack in specifying the limitations of the structure in term of working conditions (dependence on frequency, distance, configuration of the measurement), that should be explained in paper.

References:

In my opinion there is a significant issue in the bibliography as it does not include any reference about spiral diffraction grating structures.

Further, the references list on acoustic metasurfaces should be expanded. Besides the reference cited

above, many metasurfaces have been recently reported that realize Fresnel-type focusing structures.

Yours sincerely,
Marco Ricci,
University of Calabria

Response to the Reviewers

We extend our appreciation to all reviewers for their thorough evaluation of our manuscript and valuable insights that significantly contribute to the refinement of our work. We thank the Reviewer #1 and Reviewer #3 for their overall positive assessments. We are greatly encouraged by their remarks that recognize the paper's intriguing and innovative nature. We would also like to acknowledge the Reviewer #2 for pointing out the places in which our paper could be strengthened. In response to their constructive feedback, we have incorporated additional content to address the raised concerns. In the following sections, we provide comprehensive responses to each of the comments that were raised. To improve readability, all modifications made to the manuscript are highlighted in red.

Response to Reviewer 1

Comment P 1.1 — *This manuscript described an acoustic meta-differentiator based on vortex pattern to enhance intensity contrast of transmitted waves for ultrasound imaging. The physical principle as claimed by the authors was due to the added wavevector of the vortex pattern from the meta-differentiator that scattered the image information to a high k component. The claims of the detection of images by transmitted waves through masks and meta-differentiator were supported by numerical and experimental studies. The manuscript is well-written, and the results are interesting. However, there are three major concerns to be addressed.*

Response: We would like to thank the reviewer for carefully reading our manuscript. The reviewer stated that this manuscript is well-written and the results are interesting. We are delighted to address the points that were raised by the reviewer.

Comment P 1.2 — *(1) A major concern I have is the lack of control experiments to provide direct evidence of the improved edge detection as claimed by the authors. All the simulations and experiments were through both the target masks and the meta-differentiator. There are no control experiments on the imaging of the target masks without transmitting through the meta-differentiator. Thus, it is difficult to know how much edge contrast enhancement the meta-differentiator achieved. Therefore, the significance of the meta-differentiator is vague.*

Response: We appreciate the reviewer for bringing this critical issue to our attention, and we concur with the reviewer's point regarding the importance of incorporating control experiments to provide direct evidence of our meta-differentiator. In accordance with the reviewer's valuable suggestion, we have performed a series of control experiments specifically focused on the imaging of the target masks without involving the meta-differentiator. This is done to emphasize the substantial contribution of our meta-differentiator in achieving improved edge detection.

Figure R1a illustrates an amplitude object engraved into a steel plate, featuring a hollow pattern resembling the numeral 2. This object possesses dimensions of 6λ in height, 3.75λ in width, and a slit measuring 0.75λ . Here, the ultrasonic transducer is operated at 1 MHz, with the acoustic wavelength of $\lambda = 1.5$ mm. In Fig. R1b, we present the simulated and measured acoustic intensity distributions

Figure R1: Control experiments on target object imaging in the presence and absence of an acoustic meta-differentiator. (a) An amplitude object crafted from a steel plate featuring a hollow pattern of the numeral 2. Simulated and measured acoustic intensity distributions at the imaging plane $z = 80\lambda$ away from the amplitude object (b) without and (c) with the employment of acoustic meta-differentiator. (d)-(f) Analogous to (a)-(c), but involving a phase object constructed from a resin plate with a hollow pattern of the numeral 2 at its center. (g)-(i) Equivalent to (d)-(f), but featuring a phase object with the complementary physical numeral 2 made of resin material.

at the imaging plane, which is located $z = 80\lambda$ away from the amplitude object. This is done without using the proposed meta-differentiator. It is observed that the configuration of the numeral 2, which has an extremely weak intensity, is hardly discernible due to sound field diffraction. As comparison, Fig. R1c displays the simulated and measured acoustic intensity distributions that have been processed by the meta-differentiator. These distributions were also captured at the measuring plane located 80λ away from the amplitude object. The meta-differentiator successfully extracts and highlights all the boundaries of the numeral 2 with a uniform intensity, underscoring its role as an image edge extractor.

In addition, we examine a phase object in the shape of the numeral 2, both with and without the use of the meta-differentiator. The phase object is prepared from a circular plate made of resin material with the diameter of 90 mm. A hollow numeral 2 is engraved at its center, with dimensions of 6λ in height, 3.75λ in width, and a slit measuring 0.75λ , as depicted in Fig. R1d. The resin plate has a thickness of 1.2 mm, which ensures complete transmission of ultrasonic waves and introduces a phase delay of $2\pi/3$ compared to the hollow region. The ultrasonic transducer emits a Gaussian plane wave with a width of 12.7 mm. In Fig. R1e, we present the simulated and measured acoustic fields at the plane located $z = 80\lambda$ away from the phase object without being processed by the meta-differentiator. It is evident that no information regarding the numeral 2 is revealed in the far-field acoustic field distribution. In contrast, Fig. R1f showcases the simulated and measured intensity profiles at the output plane of the imaging system with the acoustic meta-differentiator. It reveals an isotropic enhancement of intensity distribution along all edges of the numeral 2. Consequently, the meta-differentiator facilitates the easy identification of phase object contours in the far field ($80\lambda \gg \text{critical distance } R^2/\lambda = 16\lambda$ with R being the radius of transducer), and aids in distinguishing objects with phase differences from the ambient background.

For further validation, we examine a complementary physical numeral 2 made of resin material, featuring a thickness of 1.2 mm and a phase advance of $2\pi/3$ relative to the ambient medium (Fig. R1g). The acoustic intensity profiles of the phase objects, with and without the assistance of the meta-differentiator, are illustrated in Figs. R1h and R1i, respectively. Without the meta-differentiator, it is challenging to distinguish the phase object from the surrounding medium. However, with the use of the acoustic meta-differentiator, all edges of the numeral 2 exhibit significant enhancements for easy recognition. Therefore, our meta-differentiator is capable of reconstructing objects in the far field and exhibits superiority in revealing intricate boundary information.

Following the reviewer's suggestion, we have incorporated the discussions regarding control experiments conducted with and without the meta-differentiator into the revised manuscript, which can be found on **Page 13**: “**Note that in the imaging system without the employment of spatial differentiator, discerning amplitude and phase objects from the surrounding medium proves challenging, owing to the unavoidable sound diffractions in the far field (refer to Supplementary Note 19 for detailed control experiments). The incorporation of acoustic meta-differentiator, however, enables the reconstruction of far-field images for both amplitude and phase objects obviating the need for post-processing reconstruction algorithms, characterized with enhanced edge features that facilitate straightforward identification.**”. Furthermore, we have included Fig. R1 and the comprehensive discussions in **Supplementary Note 19**, accompanied by **Figure S18**.

Comment P 1.3 — (2) *Another major concern is about the potential of the technology in practical applications of ultrasound imaging. In the clinical setting, the ultrasound machines reconstruct*

images based on backscattering of waves from impedance changing surfaces. This is true for A mode, B mode, and M mode imaging. The meta-differentiator proposed by the authors is based on the transmitted waves, which is usually used in bright field imaging of optical microscopy and X-ray imaging. Thus, it is hard to image how the proposed technique can be applied for practical ultrasound imaging, which dramatically reduces the significance of the work.

Response: We thank the reviewer for providing us with the opportunity to address this pertinent issue. The reviewer's concern is specifically focused on the potential application of this technology in the field of ultrasound imaging. In this regard, we wish to convince the reviewer that our proposed technology holds potential applications from two perspectives.

(i) Potential of Transmission-mode Meta-differentiator

We acknowledge the reviewer's observation that our proposed meta-differentiator relies on the analysis of transmitted waves, which is similar to the principles underlying bright field imaging in optical microscopy and X-ray imaging. We also agree with the reviewer that in clinical settings, ultrasound machines primarily reconstruct images by analyzing wave backscattering from surfaces with varying impedance. This principle applies to A mode, B mode, and M mode imaging. Within these reflective-mode imaging, the information embedded in reflected waves unveils intricate details of detected objects, with both intensity and phase serving as discerning indicators for various materials and defects. The time-of-flight measurement of the pulses reflected from each surface further imparts critical depth information for the sample. Despite the advantages of reflective-mode imaging, such as high resolution and detection accuracy, the exhaustive identification of defects across different layers of the sample proves to be time-consuming [*Applied Microscopy* 50, 1-11 (2020)]. Conversely, transmitted-mode imaging predominantly finds utility in the rapid and straightforward screening of internal defects within samples, especially in scenarios involving the assessment of a large quantity of specimens. It proves to be more expedient than reflective mode, facilitating the convenient determination of the presence of defects within the material under test. However, it falls short in precisely determining the depth location of internal defects during the material testing process. It is important to note that, in addition to the traditional machines that deal with reflected ultrasound, relevant theoretical and experimental studies have shown the possibility of achieving ultrasonic imaging with transmitted ultrasound, thereby providing additional attenuation and phase information within the sample [*Acoustic microscopy*[M], Oxford University Press, (2010); *Science* 188, 905-911 (1975)]. Consequently, advancements in the field have led to the development of **acoustic microscopes** and **ultrasound computed tomography** in the transmission mode, mirroring the progress seen in optical microscopy and X-ray imaging.

For example, **acoustic microscopes** has been developed to investigate the internal structure of materials using high-frequency ultrasound waves [*Acoustic microscopy: Fundamentals and applications*[M], John Wiley & Sons, (2008); *Science* 188, 905-911 (1975), *Appl. Phys. Lett.* 24, 163-165 (1974); *Appl. Phys. Lett.* 31, 317-320 (1977); *Proc. Natl. Acad. Sci. U.S.A* 78, 1656-1660 (1981)] in both transmitted-mode and reflected-mode. This device offers a distinct advantage in providing valuable insights into material properties, including stiffness, density, and acoustic impedance, achieved through the analysis of sound speed and attenuation within the sample. Notably, it also possesses the unique capability to visualize subsurface structures and features within transparent or semi-transparent materials that are beyond the reach of optical microscopes. In the domain of biological and biomedical research, acoustic microscopes plays a pivotal role in enabling the visualization and analytical study of biological tissues, cells, and biomaterials, enhancing our understanding of tissue architecture and cellular behaviors [*Science* 188, 905-911 (1975); *Sensors* 23(4), 1916 (2023); *Ultrasonics* 99, 105949 (2019)].

Additionally, acoustic microscopes assumes a crucial position in the field of microelectronics packaging, facilitating the comprehensive assessment of the structural integrity of microelectronic components, including wire bonds, solder joints, and die-attach quality [Adv. Mater. 5, 508-519 (1993); Rev. Mod. Phys. 67, 863 (1995); Energy Technol. 2300323 (2023)].

On the other hand, the emergence of **ultrasonic computed tomography** (ultrasonic CT) parallels the concept of X-ray computed tomography (X-ray CT) [Quantitative Ultrasound in Soft Tissues (Dordrecht: Springer) (2013); Front. Mater. 4, 40 (2017); Acoust. Imag. 28, 223–229 (2007)]. While X-ray CT employs ionizing radiation to create images of the body's internal structures, ultrasonic CT achieves a similar objective using non-ionizing ultrasound waves to ensure a safer imaging modality. The ultrasonic CT's non-ionizing ultrasound waves are considered safe and suitable for imaging, especially for vulnerable populations such as pregnant women, children, and individuals requiring frequent imaging. Additionally, ultrasonic CT excels in providing excellent soft tissue contrast, making it particularly valuable for imaging abdominal organs, the cardiovascular system, and soft tissue structures such as muscles and tendons. Moreover, ultrasonic CT offers functional imaging capabilities, enabling healthcare professionals to observe dynamic processes within the body, including organ movement, blood flow, and fetal development. From cost perspective, ultrasonic CT is generally more cost-effective, with lower equipment and maintenance costs, coupled with the absence of ongoing expenses related to radiation protection and shielding.

Building upon the advancements in acoustic microscope and ultrasonic CT technology, our **meta-differentiator** can introduce an **edge-enhanced imaging technique as a valuable compensation for conventional bright-field imaging**. This innovative approach accentuates and improves the visibility of object boundaries and regional distinctions within images. In the context of medical imaging, edge enhancement holds the potential to enhance the visualization of anatomical structures and abnormalities, thereby contributing to the early detection of diseases, aiding surgical planning, and enhancing the interpretation of medical imagery. In industrial applications, edge-enhanced imaging also serves as an essential tool for identifying defects, cracks, and irregularities in manufactured products, thereby improving quality control and inspection procedures. Conclusively, our meta-differentiator has the potential to be integrated with acoustic microscopes or ultrasound CT techniques to achieve superior object recognition capabilities. We fully agree with the reviewer that the practical implementation of these new techniques for ultrasound imaging is yet to be carried out, particularly in terms of clinical verification, and we are diligently pursuing this ambitious objective.

(ii) Demonstration of Reflected-mode Meta-differentiator

Building upon the thought-provoking comment from the reviewer and in addition to the transmission-mode meta-differentiator, we further introduce a reflective-mode acoustic meta-differentiator as illustrated in Fig. R2a. In this configuration, ultrasonic waves pass through an object situated at the plane $z = -f$, subsequently impinge upon the reflected-mode meta-differentiator positioned at $z = 0$, and ultimately undergo reflection, culminating in the formation of an image with enhanced edge delineation at the incident plane $z = -f$. The amplitude and phase meta-gratings applied to the reflective-mode meta-differentiator are depicted in Figs. R2b and R2c, respectively. Differing from the transmission-mode setup, the resin height distributions on the reflected-mode phase grating are halved due to the doubled reflected acoustic distance, and are set as 0.15λ and 0.75λ , respectively, to ensure π differences between two reflected phase steps. Meanwhile, the reflective-mode amplitude modulator is entirely inverted compared to its transmission-mode counterpart, with decreased sound transmittivity but increased reflectivity along the radial direction, in accordance with the Eq. (6) in the manuscript. In Fig.

Figure R2: Reflective-mode meta-differentiator for edge enhancement of amplitude objects. (a) Schematic illustration of employing a reflective-mode meta-differentiator in the edge-enhanced imaging system. (b) Discrete phase modulations on the reflective meta-differentiator corresponding to the distribution of resin thickness. (c) Discrete amplitude modulations represented by the distributions of slit width. (d) An amplitude object featuring a small circular hole with a radius of 0.5λ at the input plane $z = -f$. (e) Simulated intensity and phase distributions of the reflected acoustic field at the image plane $z = -f$. (f)-(g) Same as (d)-(e), but featuring an amplitude object in shape of the hollow numeral 2 with dimensions of 6λ in height, 3.75λ in width, and a slit measuring 0.75λ .

R2d, we introduce an amplitude object featuring a small circular hole with a radius of 0.5λ at the input plane $z = -f$. This object serves to confirm the point spread function (PSF) of the reflective-mode meta-differentiator in the imaging system. Figure R2e presents simulated acoustic intensity and phase distributions of the reflected field at the image plane $z = -f$. It should be emphasized that the incident waves are continuous rather than impulse waves, thus the reflected field is obtained by subtracting the incident waves from the total field. It is observed that the PSF of reflective-mode meta-differentiator manifests as a high-intensity doughnut pattern accompanied by a spiral phase carrying a first-order topological charge, akin to that of the transmission-mode counterpart. Furthermore, we examine the case of an amplitude object shaped like the numeral 2, with dimensions of 6λ in height, 3.75λ in width, and a slit measuring 0.75λ , as illustrated in Fig. R2f. The corresponding acoustic intensity and phase profiles of reflected fields are depicted in Fig. R2g, wherein all edges of the numeral 2 are prominently highlighted with isotropic intensity distributions. Hence, the reflective-mode meta-differentiator demonstrates outstanding performance in edge extraction capabilities as well. We also investigate the case where the meta-differentiator is not used and observe that sound diffractions impose limitations on the reconstruction of the object image in the far field. These limitations are similar to the fields depicted in Fig. R1(b), which are not shown here.

Finally, in addition to the two perspectives discussed above, we also propose an imaging methodology based on backscattering of waves, that relies on the utilization of an acoustic transmission-mode meta-differentiator, specifically designed for pragmatic deployment in clinical settings (refer to Fig. R3a for visual representation). In the simulation, the background sound field emits Gaussian acoustic waves with the radius of 4λ directed towards an object situated at plane $z = f$. The subsequently reflected scattering waves then traverse through the transmission-mode meta-differentiator positioned at $z = 0$ and ultimately coalesce to form an image characterized by edge enhancements at imaging plane $z = -f$. An amplitude object featuring a small circle with a radius of 0.5λ is placed at the object plane $z = f$, as depicted in Fig. R3b. The majority of acoustic waves bypass the small circular object, while a fraction of them are reflected to form an image of circle. After post-meta-differentiator processing, the simulated intensity and phase distributions of the acoustic field at image plane $z = -f$ are presented in Fig. R3c. Notably, the scattered PSF of the meta-differentiator exhibits a high-intensity doughnut pattern accompanied by a spiral phase carrying a first-order topological charge, consistent with the findings detailed in the manuscript. Furthermore, we consider the case of an amplitude object shaped like the numeral 2, with dimensions of 6λ in height, 3.75λ in width, and a line thickness measuring 0.75λ , as depicted in Fig. R3d. The corresponding intensity and phase profiles for the scattered waves processed by the meta-differentiator are displayed in Fig. R3e. It is observed that all edges of the numeral 2 are accentuated with isotropic intensity distributions, affirming the efficacy of the meta-differentiator in edge detection of scattered waves. We also investigate the scenario in which the proposed meta-differentiator is not employed, which is not presented in this section. It is observed that the reconstruction of object image in the far field is impeded due to sound diffractions. Conclusively, our proposed meta-differentiator serves as a robust image processing tool for the edge detection of both transmitted and scattered fields. A subsequent consideration pertains to the transition from continuous waves to pulse waves within the reflective-mode imaging scheme, a shift with promising implications for the practical utilization of the proposed meta-differentiator in clinical settings for medical diagnosis.

Following the reviewer's suggestion, we emphasize this issue in the revised manuscript on **Page 11**: "Furthermore, our meta-differentiator serving as an image processor can incorporate principles applicable

Figure R3: Reflective-mode edge-enhanced imaging utilizing an acoustic meta-differentiator. (a) Schematic representation of reflective-mode edge-enhanced imaging employing an acoustic meta-differentiator within the imaging system. (b) Amplitude object in the shape of a small circle with a radius of 0.5λ . (c) Simulated intensity and phase distributions of the acoustic field at the image plane. (d)-(e) Equivalent to (b)-(c), but featuring an amplitude object in the form of the numeral 2 with dimensions of 6λ in height, 3.75λ in width, and a line thickness measuring 0.75λ .

to both transmission- and reflective-modes, enabling augmented observational capacity through the comprehensive extraction of edge information from objects (see Supplementary Note 16). Through a strategic rearrangement of incident waves and detected objects, the transmission-mode meta-differentiator also can effectively process backscattering waves from objects, extracting all object edges, as elaborated in Supplementary Note 17.”. We also have provided detailed content related to this discussion in **Supplementary Notes 16 and 17**, along with **Figures S15 and S16**.

Additionally, we have included discussions regarding the practical applications of this technology within the domain of ultrasound imaging on **Page 17**: “Building upon the advancements in acoustic microscope and ultrasound CT technology, our meta-differentiator introduces an edge-enhanced imaging technique as a valuable complement to conventional bright-field imaging. This innovative approach, which improves the visibility of object boundaries and accentuates regional distinctions within images, can find applications in biomedical imaging and nondestructive testing. In the context of medical imaging, edge enhancement holds significant potential for enhancing the visualization of anatomical structures and abnormalities, thereby contributing to the early detection of diseases, facilitating surgical planning, and augmenting the interpretation of medical imagery. In industrial applications, edge-enhanced imaging can serve as an indispensable tool for identifying defects, cracks, and irregularities in manufactured products, thereby enhancing quality control and inspection procedures. Conclusively, our meta-differentiator is promising to be seamlessly integrated with acoustic microscopes or ultrasound CT techniques to deliver superior object recognition capabilities.”.

Comment P 1.4 — (3) *There is also a concern about the physical principle of the meta-differentiator. While sharp edges lead to high k components in the scattered wave, the proposed technique of further increasing the wavevector magnitude has a risk of scattering imaging information to near-field components, and thus affecting the resolution or distort the image. This needs to be discussed.*

Response: The reviewer has raised a critical concern regarding the potential risk of increased wavevector magnitude, which may result in the scattering of imaging information to near-field components. We agree with the reviewer’s point and have conducted additional analyses to elucidate the impact of the heightened tangential wavevector (specifically, the topological charge (TC) of the spiral phase) on the propagation of acoustic fields.

Figure R4a illustrates the axial acoustic field distribution of PSF along the propagation direction (left panel), corresponding to the case of zero tangential wavevector. In this configuration, the radius of point source is 0.5λ , the radius of acoustic metasurface is 30λ , and the focus position is set as 40λ . It is noteworthy that the focusing spot precisely resides at $z = 40\lambda$ with the focal length of about 8λ . In the right panel, the acoustic fields extracted from the focusing plane correspond to an Airy-like spot with an annular phase distribution, resulting in no phase modulation along the azimuthal direction. Upon introducing the first-order spiral phase into the acoustic metasurface, the axial and transverse acoustic field profiles are depicted in the left and right panels of Fig. R4b, respectively. It is evident that the additional tangential wavevector induced by the spiral phase with a TC of 1 leads to an extremely minor decrease in the axial wavevector along the propagation axis, which can be considered negligible. From the transverse acoustic fields, we observe the presence of a spiral phase carrying orbital angular momentum of 1 together with an annular intensity profile. In Fig. R4c, we extend our analysis to the case with a spiral phase of 10. Here, we find that the focal position is reduced to about $z = 39\lambda$ but

Figure R4: Propagation characteristics of the acoustic field of PSF with the increased tangential wavevector. (a) Left panel: axial acoustic intensity distribution along the propagation direction without a tangential wavevector. Right panel: corresponding acoustic intensity and phase profiles extracted from the focal plane as denoted by the yellow dashed line. (b), (c) Equivalent to (a) but incorporating additional tangential wavevectors induced by the spiral phases with TCs of 1 and 10, respectively.

still located at the distance far away from the metasurface, and the transverse acoustic field corresponds to the vortex carrying TC of 10. Therefore, the heightened tangential wavevector can contribute to a reduction in the focus position, but the acoustic fields still propagate over an extensive distance.

It is worth noting that, in this manuscript, the meta-differentiator operates based on a physical principle represented by the transfer function $t_{\text{tran}}(x', y') \propto i(x' \pm iy')$ (specifically, featuring a spiral phase with a TC of ± 1). This transfer function enables the meta-differentiator to perform 2D differentiation operations on objects, resulting in isotropic edge enhancements. Higher-order spiral phases with elevated tangential wavevector components (represented as k_t) are not effective in extracting complete edge information from objects, but can be employed for other specialized purposes [Twisted Photons: Applications of Light with Orbital Angular Momentum, Wiley-VCH, Weinheim, Germany, (2011)].

Following the reviewer's suggestion, we have incorporated a concise discussion addressing this point in the manuscript on **Page 8**: "It is found that only the transfer function characterized by $\pm 1^{\text{st}}$ -order spiral phases is capable of performing 2D differentiation operations on objects for isotropic edge enhancements. Higher-order spiral phases with increased tangential wavevector components are not effective in extracting complete edge information from objects, but could be employed for other specialized purposes [25].".

Comment P 1.5 — (4) Besides the three major concerns, there are several other minor issues to be addressed, like the "biomedical and engineering" in the first sentence of the abstract should be biomedical engineering.

Response: We appreciate the reviewer for bringing up this issue. We have made the revision by changing "biomedical and engineering" to "biomedical engineering" in the abstract. Additionally, we

have addressed and refined other minor issues throughout the manuscript to enhance its clarity and readability.

Response to Reviewer 2

Comment P 2.1 — *The authors proposed using acoustic metasurfaces to enhance isotropic edge boundary imaging in immersion. The method employs amplitude and phase modulation to highlight object features. This work can prove to be a gateway for the practical applications of using it to combat contrast issue. However, since the approach utilizing amplitude and phase modulation has already been theoretically explored in existing studies, this research is not entirely novel. Additionally, the absence of comparative analysis with various approaches to edge boundary detection makes it challenging to consider this paper for publication. Furthermore, one of the primary reasons is the complete absence of proposals regarding potential applicability to existing ultrasound imaging equipment. Nevertheless, considering that the paper effectively substantiates its theoretical foundation with experimental support and holds several implications of significance, I highly recommend transferring this work to another journal.*

Response: We sincerely appreciate the reviewer for his/her meticulous review of our manuscript. Having thoroughly read the reviewer's report, we are delighted to learn that our work is regarded as a promising gateway for practical applications aimed at addressing contrast issues. However, there are certain critical aspects that have not been sufficiently elucidated, causing the reviewer to question the suitability of this work for publication in this journal. The primary concerns raised by the reviewer are as follows:

1. Justification of the novelty of our research.
2. Comparative analysis with various approaches.
3. Explanation of the potential applicability to existing ultrasound imaging equipment.

We are eager to address these specific points and questions raised by the reviewer, and we have ensured that our comprehensive response is provided in the upcoming sections.

Comment P 2.2 — *Some of the following details also should be addressed to be published in the other journal.*

Comment 1 : A comprehensive account of the production of the suggested metasurface for experimental validation is absent. It is imperative to elucidate the specifics concerning the material, manufacturing technique, and production process to a degree that allows readers to replicate it accurately.

Response: We thank the reviewer for drawing attention to this fundamental matter. In light of the reviewer's valuable suggestion, we have offered comprehensive clarification regarding the selection of materials, manufacturing techniques, and the production process of the experimental samples. This refinement guarantees that readers can obtain a thorough understanding of our research and accurately replicate the experiments.

Figure 3c in the main text displays a photograph of the fabricated phase meta-grating, composed of two components crafted from photosensitive resin and stainless steel. The scale bar in the figure represents a length of 22.5 mm. Due to current technological constraints, the fabrication process of the

phase meta-grating is divided into two separate steps. Initially, the resin-based component is manufactured using stereolithography-based 3D printing technology (microArchR S240, projection micro stereolithography; resolution $10\ \mu\text{m}$; BMF Precision Tech Inc., <https://www.bmftec.cn/3d-printers/s240>), a well-established method in advanced material manufacturing, widely applied in the fabrication of acoustic metamaterials. Stereolithography employs photosensitive resin that can be solidified through exposure to UV laser. The photosensitive resin material boasts a mass density of $\rho_r = 1176\ \text{kg m}^{-3}$ and a longitudinal sound speed of $c_r = 2460\ \text{m s}^{-1}$, making it a highly regarded substance for modulating the transmittance and phase of ultrasonic waves within water medium. A precisely controlled UV laser follows a predetermined path and directs its beams into the resin reservoir, then the photocurable resin polymerizes into a 2D patterned layer. After each layer solidifies, the platform descends and another layer of uncured resin is prepared for patterning. Upon completion of the printing process, all printed components undergo post-curing to attain maximum strength and stability. Additionally, any rough surfaces with unevenness are meticulously sanded to achieve a smooth finish. The print resolution of the resin-based components measures approximately $10\ \mu\text{m}$, and the surface roughness is reduced to about $R_a = 3.2\ \mu\text{m}$ through post-processing. Next, the steel-based component, made of stainless steel sheet with a thickness of $200\ \mu\text{m}$, is strategically inserted into the interspace between neighboring resin units to minimize undesired sound interactions.

Concerning the amplitude meta-grating, conventional laser-cutting techniques have proven inadequate in meeting the requisite resolution standards. As an alternative approach, we have turned to 3D metal printing technology (EOS M 100, direct metal laser sintering; resolution $100\ \mu\text{m}$; GuangZhouXinYuan Metals&TCO., <http://xy-metal.com>), specifically powder bed fusion, to fabricate the amplitude meta-grating using stainless steel. The selected material possesses a mass density of $\rho_s = 7800\ \text{kg m}^{-3}$, along with longitudinal and transverse sound speeds of $c_{sl} = 6100\ \text{m s}^{-1}$ and $c_{st} = 3300\ \text{m s}^{-1}$, respectively. This additive manufacturing method entails the use of a precisely controlled laser beam, which scans the steel powders along a predetermined path and sinters them through localized heating. Under the influence of high-powered lasers, neighboring powder particles fuse together via molecular diffusion, enabling the subsequent layering process to commence. Upon the completion of the amplitude meta-grating's construction, additional optimization steps are undertaken. These include post-processing treatments like heat detachment from the build plate, accomplished through techniques such as cutting, machining, hot isostatic pressing, or solution annealing, all of which serve to enhance the overall quality of the fabricated components. As a finishing step, the roughness of amplitude meta-grating is further refined to about $R_a = 6.3\ \mu\text{m}$ through grinding and polishing procedures.

We have included the process of fabricating and characterizing the meta-differentiator sample in **Method** section on **Page 18**: “**Sample fabrication.** The fabricated phase meta-grating was composed of two components crafted from photosensitive resin and stainless steel. Due to current technological constraints, the fabrication process of the phase meta-grating was divided into two separate steps. Initially, the resin-based component was manufactured using stereolithography-based 3D printing technology (microArchR S240, projection micro stereolithography; resolution $10\ \mu\text{m}$; BMF Precision Tech Inc., <https://www.bmftec.cn/3d-printers/s240>). Stereolithography employed photosensitive resin material that can be solidified through exposure to UV laser. Then, a precisely controlled UV laser directed its beams into the resin reservoir along a predetermined path to polymerize the photocurable resin into a 2D patterned layer. After each layer solidified, the platform descended and another layer of uncured resin was prepared for patterning. Upon completion of the printing process, all printed components undergone post-curing to attain maximum strength and stability. Additionally, rough surfaces with unevenness

were meticulously sanded to achieve a smooth finish with the roughness of about $R_a = 3.2 \mu\text{m}$. Next, the steel-based component, made of stainless steel sheet with a thickness of $200 \mu\text{m}$, was strategically inserted into the interspace between neighboring resin units to minimize undesired sound interactions.

The amplitude meta-grating made of stainless steel was fabricated through 3D metal printing technology, specifically powder bed fusion (EOS M 100, direct metal laser sintering; resolution $100 \mu\text{m}$; GuangZhouXinYuan Metals&TCO., <http://xy-metal.com>). This additive manufacturing method used a precisely controlled laser beam to scan the steel powders along a predetermined path and sinter them through localized heating. Under the influence of high-powered lasers, neighboring powder particles fused together via molecular diffusion, enabling the subsequent layering process to commence. Upon the completion of amplitude meta-grating, post-processing treatments were taken such as cutting, machining, hot isostatic pressing, or solution annealing, all of which served to enhance the overall quality of the fabricated components. As a finishing step, the roughness of amplitude meta-grating was refined to about $R_a = 6.3 \mu\text{m}$ through the grinding and polishing procedures." We believe that these additional details and discussions effectively address the reviewer's concerns and provide clarity on this aspect of our work.

Comment P 2.3 — *Comment 2 : In this paper, the theoretical exposition of enacting the suggested concept appears rather robust. However, the methodology proposed herein does not introduce novel theories but rather elucidates pre-existing ones, thus leaving the extent of advancements in comparison to established literature somewhat unclear. Moreover, though experimental implementation and validation were undertaken, the account of experimental intricacies remains notably inadequate. Especially concerning metamaterials, as the practical realization holds more significance and complexity than theoretical advancement, a more explicit and detailed elaboration of the metasurface is imperative to effectively manifest the research team's claimed theory.*

Response: We appreciate the reviewer for highlighting the important methodological issue in our work. In alignment with the prevailing literature, we construct an acoustic meta-differentiator incorporating both amplitude and phase modulations, which is inherent and unavoidable as inferred from theoretical analysis. Distinguishing itself, our design theoretically features a 2D transfer function imparted with first-order spiral phase—marking a significant departure from prior methodologies [*Appl. Phys. Lett.* 110, 011904 (2017); *J. Appl. Phys.* 123, 091704 (2018)]. Meanwhile, our work pioneers the integration of acoustic orbital angular momentum into the domain of edge-enhanced imaging, extending beyond conventional applications in communication systems and particle manipulation [*Proc. Natl. Acad. Sci. U.S.A.* 114, 7250-7253 (2017); *Phys. Rev. Lett.* 121, 074301 (2018)]. Furthermore, our meta-differentiator offers distinct advantages over the existing computational acoustic metamaterials in several key aspects.

The majority of present acoustic computational metamaterials designed for differentiation operations rely on either the Fourier spatial filtering approach or the Green's function method. While these methods offer potential benefits in enhancing acoustic contrast, they do come with certain limitations that constrain their widespread applicability.

First and foremost, most current demonstrations of spatial differentiation are confined to 1D scenarios, allowing for the enhancement of only one side edge of an object along either the x or y direction. This leads to anisotropic edge enhancements that may not be optimal for imaging applications. Achieving 2D edge enhancements, which reveal both horizontal and vertical edges simultaneously, necessitates the superposition of 1D edge-enhanced images in both the x and y directions. In contrast, **our method**

inherently functions as a 2D spatial differentiator, a novel feature not reported previously in acoustics, enabling the simultaneous extraction of all object boundaries.

Moreover, the Fourier approach for spatial differentiation typically employs a planar metasurface in the Fourier domain for spatial filtering operations, along with two additional metasurfaces for applying Fourier transforms on the input and output signals. This leads to a significant increase in the overall size and complexity of computing systems, thereby impeding their suitability for compact architectures. In contrast, our proposed method utilizes just one planar metamaterial to conduct differentiation operations on the input signals. This significant reduction in the overall dimensions of the computing system opens up the possibility of compatibility with compact integrated systems.

Lastly, prevailing research predominantly focuses on edge detection of amplitude objects in air, with limited exploration of edge detection in liquid environments. However, considering the growing applications of ultrasonic imaging in medical diagnostics, where tissues and organs are often treated as phase objects in liquid, the detection of phase object edges has remained relatively unexplored in existing literature of acoustic computational metamaterials. Our work addresses this gap by providing a methodology for efficiently extracting edge information from both amplitude and phase objects in water. This advancement holds profound implications for advanced medical imaging applications.

Consequently, the novel contributions and distinct advantages of our proposed methodology stand out in the context of existing literature and pave the way for applications in acoustic fields.

Furthermore, we appreciate the reviewer for emphasizing the need for a more comprehensive experimental validation. To address this problem, we have provided additional intricate details regarding the experimental methodology and the meticulous preparation of the experimental samples.

We have emphasized the novelty and merits of our meta-differentiator compared with the previous literature on **Page 7**: “It is noteworthy that the 2D meta-differentiator exhibits a similar PSF to that of the spiral diffraction gratings [48-53]. When compared with these methods for high-order Bessel beams [48,49], focused vortex beams [50], and perfect vortex beams [51], our meta-differentiator distinguishes itself through its distinctive amplitude and phase modulation characteristics. In addition, this meta-differentiator constitutes a groundbreaking endeavor by integrating orbital angular momentum into the realm of edge-enhanced imaging, an area heretofore unexplored within the field of acoustics, extending beyond conventional applications in communication systems and particle manipulation [54,55].”, and on **Page 16**: “Importantly, our 2D meta-differentiator offers distinct advantages over the existing computational acoustic metamaterials in several key aspects (Supplementary Note 25). Primarily, this meta-differentiator with inherent 2D spatial differentiation operations can realize the simultaneous extraction of all object boundaries at the imaging plane directly, without post-processing procedures like point-to-point scanning and spatial spectrum superposition [45,46]. Meanwhile, this design leverages a single planar metastructure, incorporated into both transmission- and reflective-mode imaging systems in free space, significantly reducing the overall system dimensions and enhancing compatibility with compact integrated systems [42,43]. Furthermore, our work provides an underwater imaging methodology for efficiently extracting edge information from both amplitude and phase objects, which holds substantial promise for the advancements in biomedicine and engineering applications.”.

In addition, we have added more intricate details about the experimental methodology and sample fabrication process on **Page 18**.

Comment P 2.4 — *Comment 3 : While the paper demonstrates the feasibility of realizing high-*

contrast ultrasonic imaging through the metasurface proposed, the results section solely offers a qualitative comparison and evaluation of simulations and experiments. A quantitative assessment of the heightened contrast and resolution achieved by the proposed approach would be pertinent in elucidating and accentuating the authors' proposition. Furthermore, to accentuate the enhancement and merit of the concept posited in this study, it is deemed essential to undertake quantitative comparisons with prior research or the state-of-the-art meta-differentiators.

Response: We agree with the reviewer's point regarding the need for a quantitative assessment of the heightened contrast and resolution using our proposed approach. In response to this valuable suggestion, we present an in-depth analysis to address and clarify the raised concerns point-by-point.

(i) Contrast and Resolution

Figure R5a shows an amplitude object in the shape of numeral 2 possessing the dimensions of 6λ in height, 3.75λ in width, and a slit measuring 0.75λ , with λ being 1.5 mm. The corresponding simulated and measured intensity profiles of acoustic field at the imaging plane are displayed in Fig. R5b. It is observed that only the edges are highlighted while the background is suppressed, implying the better identification of the object. Figure R5c shows the line-scanning intensity profiles extracted from the yellow dashed lines denoted in Fig. R5b, which show the clear edge with a high contrast to the surrounding background. For quantitative analysis, the contrast is defined as the ratio of the edge intensity to the background intensity (the background intensity is evaluated from the average intensity encircled within the yellow squares in Fig. R5b) [*Light Sci. & Appl.* 11, 62 (2022)], and the full width at half maximum (FWHM) of the peak is employed to describe the width of edges [*Nat. Commun.* 6, 8037 (2015)]. Therefore, in the simulated results, the contrast of the identified edge to the background is calculated as 10~16, and the FWHM of the peak is about 0.65λ . The corresponding experimental results of the amplitude object show the contrast and FWHM of about 10~16 and 0.73λ , respectively. The measured and simulated results match well with each other, which confirms our meta-differentiator with excellent performance in the edge detection of objects.

Furthermore, we have addressed the resolution of edge detection achieved by our proposed meta-differentiator. In Fig. R6a, the object images are depicted, comprising two identical rectangular objects with a length of 5λ , a width of 0.75λ , and separation d_s of λ , 0.8λ , 0.75λ , and 0.7λ . In Fig. R6b, the upper panel illustrates the corresponding acoustic intensity distributions at the imaging plane, while the lower panel presents the line-scanning intensity along the horizontal axis indicated by yellow dashed

Figure R5: Contrast and FWHM of edge-enhanced images. (a) An amplitude object in the form of the numeral 2 with dimensions of height 6λ , width 3.75λ , and a slit measuring 0.75λ . (b) Simulated (left) and measured (right) acoustic intensity profiles at the imaging plane. (c) Line-scanning intensity profiles extracted from the yellow dashed lines indicated in panel (b).

Figure R6: Resolution of edge-enhanced images. (a) Amplitude objects including two identical rectangles with the dimensions of length 5λ , width 0.75λ , and the separation d_s ranging between λ , 0.8λ , 0.75λ and 0.7λ . (b) Upper panel: calculated acoustic intensity profiles at the imaging plane. Lower panel: line-scanning intensity profiles extracted from the yellow dashed lines indicated in upper panel.

lines. The intensity profile is normalized with respect to the maximum value. Notably, when the separation d_s is λ , all edges of the two rectangles are clearly discernible from the intensity profile. However, as we reduce the separation d_s to 0.8λ or 0.75λ , the edges of both rectangles remain observable, albeit with a slight increase in sidelobe intensity. Upon further reducing the separation d_s to 0.7λ , the intensified sidelobe intensity significantly impacts the edge extraction of inner edges between the two rectangles, rendering it challenging to distinguish between the individual objects. Therefore, the resolution of our meta-differentiator can be conservatively estimated as approximately 0.75λ . We have added the statements regarding the quantitative contrast and resolution of processed images based on our meta-differentiator in the manuscript.

(ii) Comparisons with Prior Researches

We have conducted quantitative comparisons with the existing researches to underscore the enhancements and advantages of our proposed acoustic meta-differentiator. We have compiled a comprehensive overview of key attributes in **Table 1**, summarizing the reported acoustic meta-differentiators found in prior literature sources [*Appl. Phys. Lett.* 110, 011904 (2017) [1]; *J. Appl. Phys.* 123, 091704 (2018) [2]; *New J. Phys.* 20, 073001 (2018) [3]; *Nat. Commun.* 6, 8037 (2015) [4]; *Nat. Commun.* 10, 204 (2019) [5]]. To facilitate comparison, we present information on the working principle, working mode, device component, working condition, working medium, target objects, imaging dimension, 2D image demonstration, resolution, experimental validation, and measurement method.

Table 1: Comparison among prior acoustic meta-differentiators

Literature	Zuo et al. ¹	Zuo et al. ²	Zangeneh-Nejad et al. ³	Molerón et al. ⁴	Ma et al. ⁵	This work
Working principle	Fourier spatial filtering	Fourier spatial filtering	Green's function	Trapped resonance	Fourier spatial filtering	Fourier spatial filtering
Working mode	Transmission	Reflective	Reflective	Transmission	Transmission	Transmission/Reflective
Device component	Three layers	Two layers	Single layer	Single layer	Two layers	Single layer
Working condition	Waveguide	Waveguide	Waveguide	Waveguide	Waveguide	Free space
Working medium	Air	Air	Air	Air	Air	Water
Wavelength (mm)	171.5	85.75	900	44.3	38.1	1.5
Target objects	Amplitude	Amplitude	Amplitude	Amplitude	Amplitude	Amplitude/Phase
Imaging dimension	1D	1D	1D	1D	1D	2D
2D imaging demonstration	No	No	Yes	Yes	No	Yes
Resolution (mm)	130	100	Not reported	10	9.5	1.125
Experimental validation	No	No	No	Yes	Yes	Yes
Measurement method	No	No	No	Point-by-point scanning	Spatial spectrum superposition	Direct imaging

The transverse analysis reveals that several pioneering spatial differentiation techniques involve the utilization of a single planar metasurface in the Fourier domain for spatial filtering, accompanied with one or two additional metasurfaces to execute Fourier transforms on the input and output signals. This configuration results in an expanded overall system, potentially limiting the feasibility of compact system integration. In contrast, our proposed method leverages a single planar metamaterial for conducting differentiation operations directly on input signals, leading to a significant reduction in overall system dimensions and enhancing compatibility with compact integrated systems. Meanwhile, our structure can be employed in both transmission- and reflective-mode imaging systems, further expanding the application scope of computational metamaterials. In addition, most metamaterials are confined to operation within waveguides, constraining their broader applicability. Instead, our structure operates in free space, rendering it more flexible and suitable for practical use.

In light of working medium, the predominant focus of existing research has primarily revolved around edge detection of amplitude objects in air, with limited exploration into the realm of edge detection within underwater environments. Given the burgeoning applications in the domain of medical diagnostics, where tissues and organs are frequently treated as phase objects in liquid medium, the detection of phase object edges remains an unexplored facet within the computational metamaterials literature. Addressing this significant gap, our work presents a methodology for the efficient extraction of edge information from both amplitude and phase objects in liquid. This advancement holds substantial promise for advanced medical imaging applications. Additionally, it is worth emphasizing that the majority of contemporary demonstrations in spatial differentiation are primarily limited to 1D scenarios, resulting in edge enhancements along either the x or y direction, thus yielding anisotropic edge enhancements that may not be ideal for imaging applications. Achieving 2D edge enhancements, which encompass both horizontal and vertical edges simultaneously, typically necessitates the superposition of 1D edge-enhanced images along both the x and y directions. In contrast, our method inherently functions as a 2D spatial differentiator, a novel feature heretofore unreported in acoustics, directly enabling the simultaneous extraction of all object boundaries.

Furthermore, considering the intricate design inherent in existing methodologies, limited efforts have been dedicated to experimental validation of computational metamaterials' performance. The extraction of edges through trapped resonances often mandates a laborious point-by-point scanning approach along both horizontal and vertical axes, resulting in a time-intensive process that lacks the ability to deliver real-time edge information. Meanwhile, employing multiple metastructures to individually capture high-frequency components of objects and subsequently reconstruct edge contours through spatial spectrum superposition proves to be a complex strategy that falls short in promptly revealing distinctive object characteristics. In contrast, our proposed method streamlines the processing of acoustic signals, enabling the direct reconstruction of edge-enhanced images at the imaging plane without the need for intricate post-processing procedures. This approach offers a timely imaging solution, seamlessly integrated into the observation of dynamic object motions, thus presenting a marked advancement in the imaging field.

Following the reviewer's valuable suggestion, we have added the discussions regarding the contrast and resolution of both amplitude and phase objects as processed by acoustic meta-differentiator on **Page 11**: "For quantitative analysis, the contrast is defined as the ratio of the edge intensity to the background intensity [36], and the full width at half maximum (FWHM) of the peak describes the edge width [45]. From the simulations, the contrast of the identified edge to the background is approximately 10~16, and the FWHM is about 0.65λ . The simulated energy conversion efficiency of 2D meta-differentiator,

calculated by the proportion of the sound energy at the imaging plane to that at the object plane, is about 0.45. Additionally, the corresponding measured results reveal the contrast and the FWHM are about $10\sim 16$ and 0.73λ , respectively, and the energy efficiency is estimated to be 0.27.”.

In addition, we have incorporated transverse comparisons of our meta-differentiator with prior researches into the revised manuscript **Page 16**: “Importantly, our 2D meta-differentiator offers distinct advantages over the existing computational acoustic metamaterials in several key aspects (Supplementary Note 25). Primarily, the meta-differentiator with inherent 2D spatial differentiation operations can realize the simultaneous extraction of all object boundaries at the imaging plane directly, without post-processing procedures like point-to-point scanning and spatial spectrum superposition [45,46]. Meanwhile, this design leverages a single planar metastructure, incorporated into both transmission- and reflective-mode imaging systems in free space, significantly reducing the overall system dimensions and enhancing compatibility with compact integrated systems [42,43]. Furthermore, our work provides an underwater imaging methodology for efficiently extracting edge information from both amplitude and phase objects, which holds substantial promise for the advancements in biomedicine and engineering applications.”. In addition, we have included **Table 1** along with the related discussions in **Supplementary Note 25 (Table 1)**.

Comment P 2.5 — *Comment 4* : For Figure 3b on page 21, you have presented the discrete phase modulations applied to the spatial differentiator, correlated with the distributions of resin thickness. It would be advisable to include a cross-sectional view or data on surface roughness to assess and authenticate your manufactured metasurface.

Response: We appreciate the reviewer for highlighting this crucial aspect. In response to the reviewer’s suggestion, we have incorporated the cross-sectional views of acoustic differentiator, illustrating both the phase and amplitude meta-gratings, as presented in Figs. R7a and R7b, respectively. The distributions of thickness and width parameters are also plotted in the corresponding panels. For the phase meta-grating, the discrete binary phase profile with a π phase difference is represented by variations in resin height (h), with thicknesses of 0.45 mm and 2.25 mm (depicted in yellow). To reduce wave interactions between neighboring resin units, a stainless steel sheet with a height of 3.75 mm and a thickness of $200\ \mu\text{m}$ (depicted in gray) is inserted between adjacent resin units. Regarding the amplitude meta-grating, the cross-sectional view reveals a stainless steel plate engraved with 14 annular slits, exhibiting varying widths (w) ranging from 0.2 mm to 2.8 mm with an interval of 0.2 mm along the radial direction.

According to the data provided by manufacturer, the surface roughness of the resin elements in the phase meta-grating measures approximately $R_a = 3.2\ \mu\text{m}$. Additionally, the amplitude meta-grating has undergone post-processing involving polishing, resulting in a surface roughness of approximately $R_a = 6.3\ \mu\text{m}$. Notably, both values are significantly smaller than the acoustic wavelength in water, which is approximately 1.5 mm at the operating frequency of 1 MHz. Furthermore, to address any doubts raised by the reviewer, we conduct numerical analysis to evaluate the impact of surface roughness on the imaging performance of the acoustic meta-differentiator. The analysis confirms that the device has a large tolerance in sample fabrication. Figure R8a displays a geometric surface of resin elements with random-appearing features chosen to characterize the surface roughness of the acoustic meta-differentiator. As an illustration, we set the specified amplitude scale factor as $50\ \mu\text{m}$ to quantify the roughness perpendicular to the geometric surface, which is over 15 times of the sample’s roughness. The selection of Gaussian and normal distributions ensures a continuous yet random variation in amplitudes with unrestricted roughness magnitude, as outlined in (source: <https://www.comsol.com/blogs/how-to-generate-random-surfaces-in-comsol-multiphysics/>). Subsequently, Fig. R8b presents the acoustic

Figure R7: Cross-sectional views of both (a) phase and (b) amplitude meta-gratings in an acoustic differentiator, with a plot of the thickness and width parameters that are implemented.

Figure R8: Influence of surface roughness on the imaging performance of acoustic meta-differentiator. (a) Geometric representation of the random-looking surface used to characterize the surface roughness of acoustic meta-differentiator. (b) Acoustic intensity and phase profiles at the imaging plane without accounting for the surface roughness. (c) Acoustic intensity and phase profiles at the imaging plane while considering the surface roughness.

intensity and phase profiles at the imaging plane, excluding considerations of surface roughness within the meta-differentiator. The amplitude object is configured in the shape of the numeral 2, with dimensions of 6λ in height, 3.75λ in width, and featuring a slit measuring 0.75λ . Evidently, all boundaries of the numeral 2 are well-defined with isotropic intensity. The simulated results, incorporating the consideration of surface roughness, are illustrated in Fig. R8c. Although the intensity experiences a slight reduction, all boundaries of the numeral 2 are accurately extracted, aligning closely with the results obtained without accounting for surface roughness. No significant difference is observed. Therefore, the impact of surface roughness on the acoustic field is deemed negligible and can be safely disregarded.

We have included the cross-sectional view of the acoustic differentiator, including both phase and amplitude meta-gratings, in the Supplementary Material (**Supplementary Note 13, Figure S12**). In addition, we have addressed the point regarding surface roughness on **Page 10**: “**Note that the surface roughness of resin elements in the phase meta-grating measures approximately $R_a = 3.2 \mu\text{m}$, and the amplitude meta-grating has undergone post-processing involving polishing with a surface roughness of approximately $R_a = 6.3 \mu\text{m}$. Both values are significantly smaller than the acoustic wavelength in water (1.5 mm), thereby the impact of roughness on the acoustic field can be safely disregarded.**”.

Comment P 2.6 — *Comment 5 : Figure S14 (b & c) depicts an isotropic representation of the finger boundary, whereas Figure S14 (d & e) does not manifest comparable isotropic attributes within the intensity profile. This variance could potentially be ascribed to variations in depth between the fingers. Considering that a considerable portion of the showcased measurements and simulations are based on planar structures, an important query emerges: Does the Meta-differentiator face obstacles when employed for three-dimensional structural imaging?*

Response: We appreciate the reviewer for raising this important issue and providing us with an opportunity to address it comprehensively. To clarify this matter, we will discuss the imaging performance of three-dimensional (3D) objects using our meta-differentiator in common scenarios.

In our imaging system configuration, we consider a 3D object composed of Slice I and Slice II, aligned with a separation distance denoted as d along the z direction, as visually represented in Fig. R9a. Slice I takes on the form of the numeral 2, possessing dimensions measuring 6λ in height, 3.75λ in width, and featuring a line width measuring 0.75λ . On the other hand, Slice II shows the distinct shape

Figure R9: Three-dimensional (3D) object imaging performance based on an acoustic meta-differentiator. (a) Schematic illustration of the 3D object imaging process in the imaging system employing an acoustic meta-differentiator. The 3D object consists of Slice I and Slice II. (b) Simulated acoustic intensity and phase distributions at the imaging plane $f_2 = 40\lambda$ with Slice I and Slice II positioned at 40λ and 43λ ($f_1 = 40\lambda$) away from the meta-differentiator, respectively. (c) Similar to (b), but Slice I and Slice II placed at 37λ and 40λ ($f_1 = 37\lambda$), respectively. (d) Analogous to (b), but Slice I and Slice II positioned at 40λ and 45λ ($f_1 = 40\lambda$), respectively. (e) Equivalent to (c), but Slice I and Slice II located at 35λ and 40λ ($f_1 = 35\lambda$), respectively.

of the numeral 5, sharing identical dimension properties as Slice I. When the distance f_1 is precisely 40λ , and the separation between these two slices is set as $d = 3\lambda$, that is, Slice I aligns precisely with the focal point of the imaging system. In Fig. R9b, we present simulated acoustic intensity profiles at the imaging plane at $f_2 = 40\lambda$. It is evident that the images of numerals 2 and 5 significantly overlap, posing challenges for the observation of 3D structures. Furthermore, when we modify the distance f_1 to 37λ and position Slice II at the focal point, the corresponding simulated results shown in Fig. R9c also display interference from both slice structures, rendering it challenging to distinguish between Slice I and Slice II. Hence, we concur with the reviewer's opinion that our meta-differentiator might face challenges when employed for structural slice imaging of 3D objects, exhibiting significant overlapping features within the focal length.

Additionally, we discuss the scenario wherein two slices are separated by a large axial distance. For illustration, we modulate f_1 to 40λ , and Slice I and Slice II are aligned and positioned at 40λ and 45λ , respectively, with an axial separation of 5λ . In Fig. R9d, we present the corresponding simulated intensity profiles at the imaging plane of $f_2 = 40\lambda$. Notably, the numeral 2 is easily discerned and faithfully reconstructed, with all edges prominently enhanced at the imaging plane. When we displace the meta-differentiator away from Slice I by a distance of $f_1 = 35\lambda$, then Slice II effectively occupies the focal position. Figure R9e illustrates the corresponding simulated results at the imaging plane. In this scenario, all edges of the numeral 5 are accurately extracted and distinguishable from the numeral 2. The boundaries of both slice structures are distinctly delineated, underscoring the robust performance of our meta-differentiator in 3D object imaging across different planes. Consequently, our meta-differentiator proves suitable for the edge detection of 3D objects with large separation between the interior slice structures along the focal length. Furthermore, through optimizing the acoustic meta-differentiator, it is possible to reduce the focal length of imaging system, holding promise for the detection of adjacent slices within 3D objects. In summary, our method is well-suited for the edge detection of 3D objects featuring sparse interior slice structures, but may have limitations when applied to 3D object imaging characterized by densely packed slice structures. Ongoing investigations are dedicated to addressing the edge detection of more complex 3D objects.

We have incorporated these discussions into the paper to provide a comprehensive assessment of the meta-differentiator's performance on 3D object imaging on **Page 14**: “We also discuss the application of this meta-differentiator in the context of imaging 3D structural objects, as detailed in Supplementary Note 24. It reveals that our method is well-suited for the edge detection of 3D objects featuring sparse interior slice structures; however, it may encounter difficulties when applied to processing 3D objects characterized by densely packed slice structures.”. We have included the above discussion into the Supplementary Material (**Supplementary Note 24, Figure S23**).

Comment P 2.7 — *Comment 6 : To ensure the wider applicability of the Meta-differentiator in biological imaging, incorporating three-dimensional structural imaging becomes crucial. Could the authors provide further insight into this facet and deliberate on prospective remedies or avenues for additional research aimed at resolving this concern?*

Response: We appreciate the reviewer for highlighting this crucial concern. As elucidated earlier, our meta-differentiator encounters challenges when applied to the task of edge detection in 3D structures. We concur with the reviewer's perspective that it is essential to outline potential solutions or avenues for further research aimed at addressing this issue. Several methods can be further explored in corporation with acoustic meta-differentiator to facilitate 3D structural imaging.

Ultrasonic Computed Tomography (UCT): UCT represents an advanced medical imaging modality that integrates ultrasound technology with the fundamental principles of computed tomography (CT) to produce cross-sectional images of the human body [*Neoplasia*, 22(12):770-777 (2020)]. This technique entails the strategic placement of multiple ultrasound transducers encircling the specific anatomical region of interest. These transducers emit high-frequency ultrasound waves into the body and meticulously capture the resulting echoes. To obtain comprehensive data from various perspectives, these transducers are rotated around the body. Subsequently, the accumulated ultrasound data are processed through intricate mathematical algorithms to reconstruct a 3D image of the internal structures within the body. The reconstructed 3D image can be presented as a series of cross-sectional slices, analogous to conventional CT or MRI scans, thereby furnishing clinicians with in-depth insights into the anatomy and pathological conditions of the imaged area.

Time of Flight (TOF): TOF is a foundational principle in ultrasonic imaging that plays a crucial role in determining the distances of tissue structures within the human body [*J. Ultrasound Med.* 29, 387 (2010); *Advances in Acoustic Microscopy*, Springer New York, 1995; *Ultrasonic Nondestructive Evaluation Systems*, Springer Cham, 2015]. A specialized transducer emits short pulses of high-frequency ultrasound waves into the body. Upon encountering tissue interfaces, such as the boundaries between muscle and bone or the surfaces of organs, a portion of the ultrasound wave reflects back toward the transducer, while the remainder continues propagating into deeper bodily regions. The transducer that initially emitted the ultrasound pulse then serves as a receiver, meticulously capturing the echoes generated from wave interactions with various tissue structures. The time interval between the pulse emission and the reception of each echo can reflect the exact distances between the transducer and tissue interfaces. By repeating this process with multiple ultrasound beams and echoes, the system meticulously constructs intricate 3D images that delineate the internal anatomical structures of the human body. These images serve as invaluable diagnostic tools, supporting the clinical community in disease detection, monitoring, and the guidance of medical interventions.

Therefore, we could expect to combine our meta-differentiator with these two advanced imaging methods to enhance the realization of 3D structural imaging. Firstly, we can integrate our acoustic meta-differentiator with the ultrasonic CT technique. This approach entails the rotation of meta-differentiator in conjunction with an ultrasonic transducer around the 3D object, allowing for the acquisition of a series of cross-sectional slices with edge enhancements from various perspectives. Subsequently, through the utilization of advanced image post-processing algorithms, the 3D object with edge-enhanced contours can be reconstructed from the accumulated ultrasound data. In addition, our acoustic meta-differentiator can be used in combination with the TOF technique. In this method, the transducer initially emits ultrasound pulses into the 3D object and subsequently captures the echoes generated from the wave interactions with the tissue structures at different depths. These echoes are processed through our meta-differentiator. By precisely measuring the amplitude and time intervals between the emission of ultrasound pulse and the reception of each echo, it becomes possible to reconstruct the 3D object with edge enhancements in distinct planes.

Although discussions of 3D objects are not a key focus of our work, we are grateful that the reviewer pointed out research directions holding significant promise for future investigations.

In accordance with the reviewer's suggestion, we have integrated these discussions into the paper to offer additional insight into potential research directions within the realm of 3D objects on **Page 14**: "To further enhance the capabilities of 3D structural imaging, our acoustic meta-differentiator is expected

to be integrated with either the modern ultrasonic computed tomography (CT) technique or the TOF method [1,14]. In the ultrasonic CT technique, the meta-differentiator together with an ultrasonic transducer is used to obtain a series of cross-sectional slices with edge enhancements from diverse perspectives, which are subsequently processed via image post-processing algorithms to reconstruct 3D objects with edge-enhanced contours. In the TOF method, the meta-differentiator is envisaged to process the resulting echoes induced from interactions of ultrasound pulses with tissue structures at varying depths. Based on the amplitude information and time intervals between the emission of ultrasonic pulses and the reception of each echoes, it becomes feasible to reconstruct the edge contours of 3D objects.”.

Comment P 2.8 — *Comment 7: There seems to be a lack of comprehensive elaboration regarding the theoretical importance of phase in comparison to amplitude for edge detection, along with its practical manifestation. In terms of quantitative analysis, is there a discernible enhancement when employing phase advancements and phase delays for material determination, as opposed to relying solely on intensities? Further details are required concerning the interpretation of phase information in both simulations and experiments, particularly addressing how the images deviate from the anticipated outcomes.*

Response: The reviewer raised an intriguing issue regarding the theoretical significance of phase versus amplitude in the context of edge detection. In light of this inspiring feedback, we have not only relied solely on the intensity profile, but also delved into harnessing phase advancements and delays in the phase profile to effectively characterize the detected objects [*Laser Photonics Rev.* 5, 81–101 (2011); *Phil. Trans. R. Soc. A* 375, 20150437 (2017); *Laser Photonics Rev.* 16, 2100357 (2022)].

Figure R10 presents the edge-enhanced images obtained by convolving different PSFs with the objects. Specifically, the amplitude object is configured in the shape of the numeral 2, with dimensions measuring 6λ in height and 3.75λ in width, featuring a slit measuring 0.75λ , with λ being 1.5 mm. In Fig. R10a, we depict the acoustic intensity and phase profiles of an ideal PSF within a $2\lambda \times 2\lambda$ area, which exhibits a doughnut-shaped intensity ring with a phase distribution ranging from 0 to 2π around the ring. In the output image shown in Fig. R10b, the phase singularity of the PSF causes sharp phase discontinuities along the numeral 2 with a π difference, resulting in the destructive interference of acoustic waves and zero intensity within the numeral 2. Conversely, the phase profile exhibits continuous variations along the edges, leading to constructive interference and allowing for the reconstruction of all edges of the numeral 2 with uniform intensity. Therefore, the edges of the numeral 2 can be discerned through the continuous phase variations in the phase profile. We also consider the case of a PSF within a larger area of $4\lambda \times 4\lambda$, as shown in Fig. R10c. Here, two intensity rings are evident in the intensity profile, with the first ring possessing significantly higher intensity than the second one's. Concurrently, the phase profile displays two annular distributions both carrying first-order spiral phases. The convoluted results depicted in Fig. R10d reveal that all edges of the numeral 2 are fully extracted, although with slight disturbances due to the second outer intensity ring in the PSF. The central part of the phase profile characterized by a wider width closely resembles the phase profile in Fig. R10b, while the outer phase featured by a narrower width attributed to the second phase ring has minimal influence on object confirmation. We also explore the scenario of a much larger PSF within an area of $18\lambda \times 18\lambda$, as presented in Figs. R10e and R10f. It is evident that the processed image of the object is primarily determined by the central intensity ring and spiral phase of the PSF, yielding results akin to those seen in Fig. R10d. Consequently, by not solely relying on the intensity profile, the edges of objects can also

Figure R10: Convolution of amplitude objects with the PSFs of imaging system. (a) Acoustic intensity and phase distributions of the PSF within $2\lambda \times 2\lambda$ region. Here, an amplitude object in the shape of the numeral 2 with dimensions of 6λ in height, 3.75λ in width, and a slit measuring 0.75λ . (b) Acoustic intensity and phase distributions of the output image calculated from the convolution of the PSF with the amplitude object. (c)-(d), (e)-(f) Equivalent to (a)-(b), but for the results based on the PSFs within the regions of $4\lambda \times 4\lambda$ and $18\lambda \times 18\lambda$, respectively.

Figure R11: Convolution of amplitude objects with the PSF of imaging system. (a) Acoustic intensity and phase distributions of the PSF within $2\lambda \times 2\lambda$ region. (b) Amplitude object in the shape of a ‘Panda’. (c) Acoustic intensity and phase distributions of the output image calculated from the convolution of the PSF with the amplitude object. (d)-(e) Equivalent to (b)-(c), but for the amplitude object in the form of a ‘Tree’. The images of ‘Panda’ and ‘Tree’ were hand-painted by Yurou Jia, who is the first author of this work. (f)-(j) Same as (a)-(e), but for the results based on the PSF within a region of $18\lambda \times 18\lambda$.

be ascertained by configuring the phase profile with a wider width and continuous variations.

Additionally, to further corroborate above results, we extend our analysis to the edge detection of more complex objects, incorporating both intensity and phase profiles. In Fig. R11a, we present the acoustic intensity and phase profiles of an ideal PSF for the imaging system, illustrating a doughnut-shaped intensity ring with a phase distribution ranging from 0 to 2π around the ring within a $2\lambda \times 2\lambda$ region. Figure R11b introduces an amplitude object in the form of a 'Panda' with dimensions of $60\lambda \times 60\lambda$, where the white region represents full transmission of acoustic waves, while the black area indicates no transmission. By leveraging the convolution process of the PSF with the amplitude object, we present the resulting acoustic intensity and phase profiles of the output image in Fig. R11c. In 'flat' regions, neighboring points exhibit doughnut rings with identical phase and amplitude characteristics, leading to destructive interference due to the π phase difference across the doughnut. Conversely, at the edges of the object, the PSF of neighboring points deviates either in intensity or phase retardation, resulting in a brightening effect along the edges. Therefore, both the intensity and phase profiles collaboratively enhance all edges of the 'Panda' against the background for clearer recognition. We further delve into the analysis of another amplitude object in the form of a 'Tree', as depicted in Fig. R11d. The corresponding intensity and phase profiles of the output image, processed via the convolution method, are presented in Fig. R11e. In this scenario, the phase profile effectively conveys edge information of objects, similar to the role played by the intensity profile. Furthermore, the non-uniformity in phase along the edges generates an apparent shadow effect, as revealed in [*Phys. Rev. Lett.*, 94, 233902 (2005)]. Additionally, we discuss the convoluted results based on the PSF within the area of $18\lambda \times 18\lambda$ with these complex objects, as shown in Figs. R11f-j. From the output images, the normalized intensity profiles clearly reveal the full edges of objects in a more distinct manner. Although the phase profile is more complex due to the multiple-ring spiral phase in the PSF, the edges can still be discerned along the phase profile with a wider width and continuous variations. Consequently, the edge information of objects can be more easily extracted from the intensity profile than the phase profile. To enhance the utility of phase profile for object recognition, optimizing the PSF of imaging system by eliminating the outer-ring phase profiles is necessary.

Following the reviewer's suggestion, we have made the above points clear in the revised manuscript on **Page 8**: "It is evident that the phase singularity of PSF causes sharp phase discontinuities along the numeral 2 with a π difference, resulting in the destructive interference of waves and zero intensity within the numeral 2. Conversely, the phase profile exhibits continuous variations along the edges, leading to constructive interference of waves and allowing for the reconstruction of all edges of the numeral 2. This clearly demonstrates the successful achievement of 2D spatial differentiation, effectively extracting isotropic edge information. Therefore, not exclusively dependent on the intensity profile, the demarcation of object boundaries can be elucidated by discerning the phase profile characterized by a broader width and continuous variations (Supplementary Note 5)". In addition, we have included the above discussion into **Supplementary Note 5** and **Figure S3**.

Comment P 2.9 — *Comment 8 : To the right-hand side of Fig. 2a, there exist four crimson extensions stemming from the central circle at 45, 135, 215, and 305 degrees. What factors contribute to their presence (e.g., simulation boundary conditions and artefacts) and what is their contextual significance? Furthermore, there is an additional red contour on the outer edge of the outer circle. What does this phase delay indicate?*

Response: The reviewer has raised a good point that needs to be clarified to avoid confusion. In Fig. 2a, the crimson extensions emanating from the central circle at 45, 135, 215, and 305 degrees represent a phase of 2π , while the neighboring mazarine regions indicate a phase of 0. In the phase profile, the phases are normalized within the range of 0 to 2π , thus rendering a 2π -phase equivalent to a 0-phase. To avoid any potential confusion, we have revised Fig. 2a to remove the sharp phase discontinuities.

Furthermore, it's important to note that the point spread function of the focus metasurface generates an Airy-like spot with an annular-ring phase profile. Consequently, the additional red contour on the outer edge of the outer circle arises from the phase component of the outer ring.

To ensure clarity and mitigate any possible confusion, we have made the necessary adjustments to Fig. 2a in the manuscript.

Comment P 2.10 — *Comment 9 : The acoustic metasurface is constructed using resin and a stainless steel sheet. It would be necessary to measure attenuation when ultrasound signals pass through them, and specific details about the absolute magnitude of the ultrasound pressure signals administered would be essential. While most intensity values have been normalised solely to arbitrary units, comparing the differences in specific absolute pressure values could offer crucial insights for future applications.*

Response: We appreciate the reviewer for highlighting this concern. We concur with the reviewer that it is crucial to consider ultrasound attenuation when acoustic waves pass through the resin and steel materials in acoustic meta-differentiator.

The experimental methodology employed to measure sound attenuation in different materials necessitates the comprehensive analysis of time-of-flight and amplitude of ultrasonic pulses as they traverse solid block specimens [*Nature* 537, 518 (2016)]. Specifically, the experiment process involves sending ultrasonic pulses through the material and measuring the pulses to travel through the material and be reflected back. Subsequently, the attenuation coefficients are determined from the reduction in amplitude of the reflected pulses [*Appl. Sci.* 10, 2230 (2020), *J. Appl. Clin. Med. Phys.* 23, e13495 (2022)]. The measured results reveal that the sound attenuation coefficients for photosensitive resin and stainless steel materials are determined as 3.55 dB cm^{-1} and 1.5 dB m^{-1} , respectively, at 1 MHz. Within our experimental sample, the sound attenuations for the 0.45 mm and 2.25 mm thick resin components are 0.16 dB and 0.80 dB, respectively, while that for the 1.25 mm thick steel element is 0.0019 dB, all of which are exceedingly small and may be considered negligible. We have incorporated a comprehensive description of the sound attenuation characteristics of both resin and steel materials to the manuscript.

In addition, the phase meta-grating is constructed from resin material with an acoustic impedance of approximately $2.89 \times 10^6 \text{ kg m}^{-2} \text{ s}^{-1}$, which closely matches that of the surrounding water medium ($1.5 \times 10^6 \text{ kg m}^{-2} \text{ s}^{-1}$). The transmissivity of acoustic waves passing through the phase modulator falls within the range of 0.8 to 1. The amplitude meta-grating is fabricated from stainless steel with an acoustic impedance of $4.75 \times 10^7 \text{ kg m}^{-2} \text{ s}^{-1}$ significantly higher than that of water, and its thickness is designed to be 1.25 mm (equivalent to one-quarter wavelength). This design ensures that acoustic waves only transmit through the slits, resulting in a transmissivity of approximately 0.5 for acoustic waves passing through the amplitude meta-grating. In the simulations, an incident acoustic field with the magnitude of 1 Pa is used to image the amplitude object of the numeral 2, and the energy efficiency, calculated by the proportion of sound energy at the imaging plane to that at the object plane, is calculated approximately as 0.45. Figures R12a and R12b represent the measured electric signals at the object and image planes, respectively. The time period denoted as T is equal to $1/f$ with $f = 1 \text{ MHz}$. In the exper-

iment, the function generator produces an electric signal of 280 mV, which undergoes amplification by a power amplifier before being input into the emitting transducer. It is observed that the collected electric signals at the object plane measure about 120 mV, while those at the imaging plane are approximately 32 mV. These measurements suggest that the energy conversion efficiency of acoustic waves passing through our meta-differentiator is approximately 0.27. The deviation between simulated and measured results is mainly ascribed to the errors in sample fabrication. It is important to note that, although the transmitted acoustic energy is subject to some reductions because of attenuation and reflections, the absolute pressure values of the resulting image can be effectively enhanced by increasing the power of the incident waves. This underscores the potential for optimizing the imaging process and improving the overall image quality. We have added the absolute pressure values of both simulated and measured intensities in the manuscript, which are anticipated to provide valuable insights for practical applications.

Figure R12: Experimental results of acoustic signals measured at the (a) object plane and (b) image plane.

We have incorporated the sound attenuation coefficients of sample materials to the manuscript on **Page 10**: “Meanwhile, the sound attenuation for the 0.45 mm and 2.25 mm thick resin elements are about 0.16 dB and 0.80 dB, respectively, while that for the 1.25 mm thick steel component is 0.0019 dB, all of which are exceedingly small and can be negligible [58,59].”. In addition, we have added specific details about the energy conversion efficiency of acoustic meta-differentiator on **Page 11**: “The simulated energy conversion efficiency of 2D meta-differentiator, calculated by the proportion of the sound energy at the imaging plane to that at the object plane, is about 0.45. Additionally, the corresponding measured results reveal the contrast and the FWHM are about 10~16 and 0.73λ , respectively, and the energy efficiency is estimated to be 0.27. The measurements and simulations exhibit good agreement, although slight deviations may be attributed to fabrication errors in the experimental sample. Despite the energy efficiency is subject to some reductions due to the reflections of meta-differentiator, the absolute intensity of the resultant image at the output plane can be effectively enhanced by augmenting the power of incident waves.”.

Comment P 2.11 — *Comment 10 : Ascertaining the results necessitates precise details about the specifications of the receiving hydrophone. Providing more specific information regarding the hydrophone employed is crucial. The model number alone might not be adequate for locating the specifications online.*

Response: According to the reviewer’s suggestion, we have added more information about the specifications of the employed hydrophones in the revised manuscript on **Page 20**: “This Hydrophone (PA

NH0500, output impedance $18\text{ pF} \pm 3\text{ pF}$; typical probe sensitivity $-250.5\text{ dB re } 1\text{V}/\mu\text{Pa}$; frequency range $0.1\text{ MHz}-20\text{ MHz}$; Precision Acoustics Ltd, UK) was coupled with submersible preamplifier and DC coupler to configure an integrated hydrophone system exhibiting an output impedance of $50\ \Omega$. For more comprehensive insights into the specifications of the needle hydrophone, further details can be referenced in (<https://www.acoustics.co.uk/product/0-5mm-needle-hydrophone/>).”.

Lastly, we wish to convince the reviewer that the revised manuscript is suitable for publication in *Nat. Commun.* based on the following reasons:

(1) Innovation in Acoustic Meta-Differentiator Design: Our method integrates a linear amplitude meta-grating with a spiral phase meta-grating, forming a 2D acoustic meta-differentiator for isotropic edge-enhanced imaging. This work also pioneers the integration of acoustic orbital angular momentum into the imaging realm, thereby expanding the utility of acoustic vortex beyond conventional applications in communication systems and particle manipulation.

(2) Distinct Advantages of Acoustic Meta-Differentiator: Our design distinguishes itself over the existing acoustic computational metamaterials in several aspects. Primarily, this acoustic meta-differentiator with inherent 2D spatial differentiation can realize the simultaneous extraction of all object boundaries, without post-processing procedures like point-to-point scanning and spatial spectrum superposition. Meanwhile, this design leverages a single planar structure, which can be incorporated into both transmission- and reflective-mode imaging systems in free space rather than confined in waveguide, significantly reducing the overall system dimension and enhancing compatibility with compact integrated systems.

(3) Contributions to Edge-Enhanced Imaging Techniques: Our work provides the first exploration of underwater edge-enhanced imaging of both amplitude and phase objects, which is directly aligned with the exigencies of ultrasonic diagnosis in liquid environments, where the bones or stones are like amplitude objects while the tissues or organs are treated as phase objects. The introduced edge-enhanced imaging technique, as a valuable complement to conventional bright-field imaging, is promising to be anticipated into acoustic microscopy and ultrasonic computed tomography technology for practical applications in the realms of biomedicine and engineering.

We hope our responses, together with the revised manuscript with a much clearer focus, may make our work worthy of *Nat. Commun.* We are also willing to address any additional concerns from the reviewer.

Response to Reviewer 3

Comment P 3.1 — *The manuscript presents and demonstrates a method to realize a “meta-differentiator for achieving isotropically high-contrast ultrasonic imaging”. The authors describe how to realize a prescribed spatial operation by modulating the amplitude and phase of an ultrasonic beam passing through a circular plate and show how a phase modulator composed of focus and spiral phases with a first-order topological charge can act as edge detector in ultrasonic imaging. In this regard, the results are innovative to my knowledge.*

Response: We gratefully thank the reviewer for the valuable and constructive comments, which have helped us a lot in improving our manuscript. We are strongly encouraged by the reviewer’s report that the results in this manuscript are innovative. We are delighted to address the items raised by the reviewer.

Comment P 3.2 — *However, I have two main comments about the proposed “metasurfaces” and the overall novelty of the results that the authors should address: 1) Structures very similar to the one described in the article are known in literature as “spiral diffraction gratings” or “spiral Fresnel zone plate”, depending on the characteristic of both amplitude and phase modulation, and have been used for generating or manipulating high-order acoustic or optical Bessel beams or to generate acoustic vortices.*

Not being exhaustive, see for instance:

a. Jiménez, N., Picó, R., Sánchez-Morcillo, V., Romero-García, V., García-Raffi, L. M., & Staliunas, K. (2016). Formation of high-order acoustic Bessel beams by spiral diffraction gratings. *Physical Review E*, 94(5), 053004

b. Jiménez, N., Romero-García, V., García-Raffi, L. M., Camarena, F., & Staliunas, K. (2018). Sharp acoustic vortex focusing by Fresnel-spiral zone plates. *Applied Physics Letters*, 112(20).

c. Jiménez-Gambín, S., Jimenez, N., Benlloch, J. M., & Camarena, F. (2019). Generating Bessel beams with broad depth-of-field by using phase-only acoustic holograms. *Scientific reports*, 9(1), 20104.

d. Vaity, P., & Rusch, L. (2015). Perfect vortex beam: Fourier transformation of a Bessel beam. *Optics letters*, 40(4), 597-600.

e. Zhang, B., & Zhao, D. (2010). Focusing properties of Fresnel zone plates with spiral phase. *Optics Express*, 18(12), 12818-12823.

The devices presented in the references mentioned above implement PSFs very close to that mapping the spot in Figure 2a in that of Figure 2b of the manuscript, i.e. the omni-directional differentiator.

For instance in refs. (a) and (b) various spiral diffraction gratings have been developed and tested to generate non-zero orbital angular momentum beam starting from standard piston-transducers (acoustic) or Gaussian beams (optics), i.e. approximating the transformation of an ideal Bessel J0 beam into a J1 beam.

Consider also that the PSFs reported in supplementary material note 6 are actually the patterns of Hermite-Gauss beams, which can be generated from a Gaussian beam with spatial derivative in

x- or y- direction and are very similar to the pattern generated with spiral diffraction gratings in [Yu, X., Trallero-Herrero, C. A., & Lei, S. (2016). Materials processing with superposed Bessel beams. Applied Surface Science, 360, 833-839.] as a superposition of different Bessel beams.

The authors should therefore include these topics/works in the review of the state of the art and elaborate on and highlight the differences and the novelty of their structure and method. Linking the manuscript results with the theory of Bessel beams can also add value to the paper, as there are many applications related to the manipulation of acoustic beams with orbital angular momentum.

Response: We sincerely appreciate the insightful comments provided by the reviewer and the references to the relevant articles [namely, *Phys. Rev. E* 94, 053004 (2016); *Appl. Phys. Lett.* 112, 204101 (2018); *Sci. Rep.* 9, 20104 (2019); *Opt. Lett.* 40, 597-600 (2015); *Opt. Express* 18, 12818-12823 (2010)]. We acknowledge the reviewer's observation that our phase modulator design in the acoustic meta-differentiator bears resemblance to concepts such as "spiral diffraction gratings" or "spiral Fresnel zone plates" in terms of structural design and point spread function. In response to this suggestion, we have included these references in our paper and have further enhanced the discussion on the novelty of our approach.

For instance, Jiménez et al. proposed an Archimedes' spiral diffraction grating for scattering plane acoustic waves into high-order acoustic Bessel beams [*Phys. Rev. E* 94, 053004 (2016)]. This method offers flexibility in changing the topological charge of generated vortex beams by varying the number of arms in the spiral diffraction grating. Furthermore, it allows high-order acoustic Bessel beams to propagate over long distances without significant diffraction. In addition, Jiménez et al. introduced the concept of the Fresnel-spiral, which combines the focusing properties of a Fresnel zone plate with the phase dislocation of spiral gratings, enabling the generation of strongly focused vortex beams [*Appl. Phys. Lett.* 112, 204101 (2018)]. Differing from pure amplitude modulation, Jiménez et al. also presented a straightforward method to generate zero-th and high-order Bessel beams with flat intensity along their axes using phase-only acoustic holograms [*Sci. Rep.* 9, 20104 (2019)]. In the field of optics, Zhang et al. proposed a Fresnel zone plate with spiral phase, in both amplitude and phase modulations, to generate acoustic vortices with integer or fractional topological charges [*Opt. Express* 18, 12818-12823 (2010)]. Furthermore, Vaity et al. presented a method for generating perfect vortex beams with controllable ring radius by utilizing the Fourier transform property of a Bessel beam [*Opt. Lett.* 40, 597-600 (2015)].

These innovative methods have demonstrated their efficiency in generating a wide range of acoustic vortex beams, including high-order Bessel beams, focused vortex beams, and perfect vortex beams, making them well-suited for practical applications spanning particle manipulation and communication technology. In a similar vein, our work introduces a phase-modulated metamaterial that combines the focusing phase with the first-order spiral phase to achieve first-order orbital angular momentum. The resulting point spread function exhibits an annular intensity ring and first-order spiral phase, which bears resemblance to those discussed in the referenced literature. However, in terms of practical application, our work pioneers **a novel direction by applying acoustic vortex beams to the edge detection of objects, an application that has not yet been explored in the field of acoustics.**

Furthermore, the reviewer pointed out that the PSFs presented in Supplementary Note 6 closely resemble the patterns of Hermite-Gauss beams, similar to the patterns generated using spiral diffraction gratings as a superposition of opposite Bessel beams [*Appl. Surf. Sci.* 360, 833-839 (2016)]. We appreciate the reviewer's kind efforts, and we have appropriately cited this relevant paper in the revised manuscript.

These referenced articles offer valuable insights into the generation of various vortex beams, aligning closely with the focus of our research. To underscore the significance of our work, we have included these papers as Refs. 48-51 in the reference list and provided in-depth discussions on this topic on **Page 7**: “It is noteworthy that the 2D meta-differentiator exhibits a similar PSF to that of the spiral diffraction gratings [48-53]. When compared with these methods for high-order Bessel beams [48,49], focused vortex beams [50], and perfect vortex beams [51], our meta-differentiator distinguishes itself through its distinctive amplitude and phase modulation characteristics. In addition, this meta-differentiator constitutes a groundbreaking endeavor by integrating orbital angular momentum into the realm of edge-enhanced imaging, an area heretofore unexplored within the field of acoustics, extending beyond conventional applications in communication systems and particle manipulation [54,55].”.

Comment P 3.3 — *According to [Assouar, B., Liang, B., Wu, Y., Li, Y., Cheng, J. C., & Jing, Y. (2018). Acoustic metasurfaces. Nature Reviews Materials, 3(12), 460-472.], it is generally agreed that “Acoustic metasurfaces are 2D materials of subwavelength thickness capable of providing non-trivial local phase shifts (or amplitude modulation) or extraordinary sound absorption. The uniqueness of metasurfaces lies in their ability to freely tailor the wave fields such that the phase and/or amplitude is fully controlled. The acoustic metasurface concept is based on arrays of subwavelength units, including (but not limited to) Helmholtz resonators, membranes and coiling-up space structures.”.*

The present structure does not exploit any resonance phenomenon, nor does it achieve sub-wavelength thickness (thickness equivalent to 2.5λ), so the term metasurface is in my opinion not adequate, while instead spiral diffraction grating or just diffraction grating seems more fitted, especially in the light of Eq.4 and its interpretation as a convolution with the Fourier transform of t_{tran} . Diffraction gratings indeed manipulate vectors in the k-space as the present structure does.

To clarify the distinction between metasurfaces and diffraction gratings, and to provide a further reference closely linked to the manuscript results, see for instance [Jiang, X., Li, Y., Liang, B., Cheng, J. C., & Zhang, L. (2016). Convert acoustic resonances to orbital angular momentum. Physical review letters, 117(3), 034301.] where a 0.5λ metasurface is used to apply a spiral phase shift with a 1st order topological charge. In this paper the metasurface applies a spiral phase shift by combining different resonating structures transforming a Gaussian beam in one having orbital angular momentum.

Response: We thank the reviewer for recommending two important papers that contribute to a deeper understanding of the terminology “acoustic metasurfaces”. Having thoroughly read these papers, we concur with the reviewer’s observation that the term “acoustic metasurface”, which implies the achievement of resonance phenomena within sub-wavelength thickness, may not be strictly applicable in our manuscript. In light of this valuable suggestion, we have made the appropriate revisions throughout the entire manuscript, replacing the term “metasurface” with “meta-grating” to ensure more accurate statements.

Comment P 3.4 — *Validity: The analytical, numerical and experimental analysis of the phase modulator performance is very accurate and clear, and the results reported both in the manuscript and in supplementary material very good.*

Response: We appreciate the reviewer’s positive feedback regarding the quality of the results presented in both the main manuscript and the supplementary material.

Comment P 3.5 — *Significance:* The use of the present structure in ultrasonic diagnostic or NDT imaging is not straightforward. As the authors considered just a single frequency excitation, it is not clear if the structure can work with different frequencies or, as for the Fresnel zone plates, the proposed structure works mostly at single frequency and fixed distance, i.e. the focus moves by changing the excitation frequency. If this is the case, the use of the present structure in ultrasonic imaging for diagnostic or NDT purposes is not feasible in most of the applications, which usually requires broad-band signals and a single-side inspection. Instead, the structure could be successfully used in through transmission imaging with narrowband excitation. This fundamental aspect should be discussed in the paper. Further, if the structure can instead work with a broadband signal maintaining the focal distance, this should be emphasized as this is one of the main goals in ultrasonic imaging and experimental results obtained by using a broadband signal should be added.

Response: We appreciate the reviewer for bringing up this important issue. In accordance with the reviewer’s suggestion, we have conducted further investigations to evaluate the performance of the acoustic meta-differentiator at various excitation frequencies.

Figure R13: Imaging performance of the acoustic meta-differentiator at various excitation frequencies. (a) Acoustic intensity distribution after transmitting through the meta-differentiator along the propagation direction at the operating frequency of 0.9 MHz (left panel). Acoustic intensity (top right panel) and phase (bottom right panel) distributions extracted from the imaging plane $z = 67.5$ mm as denoted by the yellow dashed line. (b)-(c) Same as (a), but for the acoustic field at the operating frequencies of 1.0 MHz and 1.1 MHz, respectively.

In the imaging system, the input object is positioned at $z = -f_1$, propagates through the acoustic meta-differentiator located at $z = 0$, and ultimately arrives at the output imaging plane situated at $z = f_2$. The meta-differentiator operates at a designated frequency of 1 MHz and possesses a specific design with a radius of 30λ (45 mm) and a focal distance of $f = 40\lambda$ (60 mm). The amplitude object, representing the numeral 2, exhibits dimensions with a height of 6λ (9 mm) and a width of 3.75λ (5.625 mm), incorporating a narrow slit measuring 0.75λ (1.125 mm) in width. When the operating frequency is adjusted to 0.9 MHz, the resulting acoustic intensity distribution transmitted through

the meta-differentiator along the propagation path is illustrated in the left panel of Fig. R13a. It is discernible that the imaging plane shifts to $f_2 = 67.5$ mm. Furthermore, the acoustic intensity and phase distributions, extracted from the imaging plane, are depicted in the top right and bottom right panels of Fig. R13a, illustrating that all edges of the numeral 2 are prominently highlighted with nearly uniform intensity. Similarly, we explore the scenario involving an operating frequency of 1.0 MHz, and the corresponding acoustic intensity and phase profiles are presented in Fig. R13b. In this case, the imaging plane shifts to $f_2 = 60$ mm, and the intensity profile at the imaging plane accurately delineates all boundary information of the numeral 2. Figure R13c showcases the acoustic intensity and phase distributions when the acoustic meta-differentiator operates at 1.1 MHz. Here, the imaging plane f_2 relocates to 54 mm, and the numeral 2 becomes distinctly discernible with all boundaries enhanced by uniform intensity. It is pertinent to note that the size of the image increases with the frequency, adhering to a similar imaging rule as optical lenses. The imaging plane of our meta-differentiator dynamically shifts with changes in the excitation frequency, which bears resemblance to the characteristics observed in Fresnel zone plates [*J. Sound Vib.* 500, 116035 (2021)]. Therefore, our proposed meta-differentiator may not be optimally suited for broadband signal processing, as its design optimization is rooted in single-frequency operation with a fixed imaging distance. However, it is crucial to emphasize that despite this inherent limitation, the meta-differentiator consistently excels in its core capability of edge detection, maintaining its effectiveness across a spectrum of frequency adjustments. Furthermore, the phenomenon of alterations in the imaging plane as a function of operational frequency exhibits potential utility in the context of imaging objects at different depths.

We have incorporated the above discussions into the paper to clarify the frequency response of acoustic meta-differentiator on **Page 9**: “**Supplementary Note 10** also reveals the performance of the 2D meta-differentiator at various excitation frequencies, where the imaging plane dynamically shifts with the variations in excitation frequency, similar to that of the Fresnel zone plates [56,57]. However, our meta-differentiator consistently excels in its core capability of edge detection across a spectrum of frequency adjustments.”. Additionally, we have included Fig. R13, along with related discussions in **Supplementary Note 10 (Figure S9)**.

Comment P 3.6 — *In respect to the edge detection of “phase-structures”, it is not true that they could be so hard to detect. By using a short pulse excitation with a broadband spectrum, such a structure can be easily detected by measuring the Time-of-flight of the ultrasonic pulse passing through the structure and then edge detection processing can be applied to the resulting image. By using narrowband signals, the phase structure can be detected as well, for instance by using the lock-in analysis that can provide a very high sensitivity in phase measurement.*

Response: We thank the reviewer for highlighting this potential source of confusion. We agree with the reviewer’s point that the edge detection of “phase-structures” can be realized by using a short pulse excitation with a broadband spectrum together with post-processing procedure. In the time-of-flight (TOF) method, a transducer emits short pulses of high-frequency ultrasound waves into the phase structures, and then serves as a receiver to capture the echoes backscattered from phase structures. The time interval between the pulse emission and the reception of each echo can reflect the exact distances between the transducer and phase structures [*J. Ultrasound Med.* 29, 387 (2010); *Advances in Acoustic Microscopy*, Springer New York, 1995; *Ultrasonic Nondestructive Evaluation Systems*, Springer Cham, 2015]. Subsequently, post-processing techniques are applied to the resultant image to obtain the objects’ edge information. However, we believe it is indeed worthwhile to point out to the readers that our

proposed meta-differentiator, functioning as an acoustic computational metamaterial, presents notable advantages through direct edge extraction in contrast to conventional post-processing approaches. This feature enhances the robustness and efficacy of our methodology across various aspects.

Low Power Consumption: Conventional digital image processing techniques demonstrate proficiency in handling intricate and extensive datasets, yet they are constrained by the limitations inherent to standard computing hardware, namely, computing speed and power consumption [*Nat. Rev. Mater.* 6, 207-225 (2021)]. In contrast, our proposed method excels in the direct processing of acoustic signals, bestowing the dual advantages of diminished power consumption and heightened processing speed. This capability empowers the execution of comprehensive data processing tasks with minimal energy expenditure.

Real-time Processing: The post-processing methodology, while adept at generating images at predefined intervals, lacks the capacity to capture the dynamic states of objects in real-time [*Nat. Rev. Mater.* 6, 207-225 (2021), *Nat. Commun.* 8, 15391 (2017)]. Conversely, our meta-differentiator offers direct imaging processing of the detected objects. This capability holds tremendous promise in applications such as the continuous monitoring of living organisms, where the prompt capture of dynamic changes and immediate insights is of paramount importance [*Optica* 5, 208-212 (2018)].

Therefore, we anticipate that our meta-differentiator, firmly rooted in the principles of acoustic computational metamaterials, may present a compelling departure from conventional post-processing techniques. It distinguishes itself through its low power consumption profile, high-speed processing capabilities, and real-time imaging capability, particularly valuable in dynamic object monitoring applications.

Furthermore, we thank the reviewer for bringing to our attention the high-sensitivity phase measurement technique employing lock-in analysis with a narrowband signal. Lock-in analysis, also known as lock-in amplification or phase-sensitive detection, constitutes a robust methodology widely employed in diverse scientific and engineering domains for extracting subtle signals from noisy backgrounds [*NDT & E International* 124, 102547 (2021); *NDT & E International* 98, 147-154 (2018); *Opt. Express* 24, 28122 (2016); *Proc. Natl. Acad. Sci. U.S.A.* 105, 17789 (2008)]. This technique involves the modulation (multiplication) of the signal of interest by a reference signal, typically a sinusoidal waveform at a known frequency termed the reference frequency. Subsequently, the modulated signal undergoes filtration through a low-pass filter to isolate solely the component oscillating at the reference frequency. Through the measurement of the amplitude and phase of this filtered signal, precise details regarding the original signal, including its amplitude and phase, are acquired. Lock-in amplifiers excel in detecting exceedingly faint signals, often submerged below the noise threshold, rendering them invaluable for scenarios where signal-to-noise ratio holds paramount importance. Furthermore, lock-in analysis furnishes not only amplitude data but also pivotal phase information about the signal, which proves indispensable for specific measurements like impedance spectroscopy or material characterization.

However, it is important to note that the application of lock-in amplifiers necessitates specialized equipment and may involve intricate setup and calibration procedures, rendering them less conducive for straightforward measurements. Additionally, lock-in analysis relies upon a stable reference signal and typically operates at a slower measurement pace compared to direct signal measurements, which could present a constraint for certain dynamic experiments. In summary, lock-in analysis stands as a potent methodology for extracting feeble signals from noise, particularly when high sensitivity and selectivity are imperative. Nevertheless, it is accompanied by considerations of complexity, cost, and the prerequisite for stable reference signals that must be carefully weighed when determining its suitability for a given experiment or application.

In contrast, our meta-differentiator in principle is primarily designed as an image processing tool aimed at extracting edge information from objects to facilitate their recognition, rather than enhancing signal visualization by mitigating noise.

As this is indeed not straightforwardly illustrated, we add the aforementioned discussions into the manuscript on **Page 12**: “**Note that the edge detection of phase structures can be achieved through the application of digital image processing techniques on the acquired image obtained from the broadband ultrasonic pulse time-of-flight (TOF) measurements [1,14]. However, our meta-differentiator, grounded in the principles of computational metamaterials, exhibits unique advantages encompassing low power consumption, high-speed processing capabilities, and real-time imaging proficiency, rendering it particularly valuable in contexts requiring dynamic object monitoring [33,34,41].**”, and “**In addition, the lock-in analysis employing narrowband signals excels in detecting phase structures as well, offering heightened sensitivity in phase measurement and serving as a robust methodology for extracting subtle signals from noise [60,61]. Despite of its advantages in scenarios where both high sensitivity and selectivity are imperative, this method is accompanied by considerations of complexity, cost, and the prerequisite for stable reference signals. In contrast, our spatial differentiator serving as an image processing tool shows the specific capability of directly extracting edge information from objects to facilitate recognition, rather than focusing on noise mitigation for signal enhancement.**”.

Comment P 3.7 — *However, the direct imaging of the edges of phase-structure is undoubtedly a very good result. Data and methodology: The methodology followed by the authors is sound, the quality of the data and presentation is very good as well. Suggested improvements: As said before, there is a lack in specifying the limitations of the structure in term of working conditions (dependence on frequency, distance, configuration of the measurement), that should be explained in paper.*

Response: The reviewer made high comments on our achievement, methodology, quality of the data and presentation. We are grateful for the reviewer’s encouraging feedback. In response to the valuable suggestion provided by the reviewer, we have taken measures to clarify the limitations of the structure, specifically concerning its operating conditions. This clarification aims to enhance the comprehensiveness of our work, and present possible trends and directions of future research.

As discussed in the responses to previous comments, we have conducted a comprehensive study on the frequency response. Here we delve into further investigations regarding the performance of the acoustic meta-differentiator with objects positioned at varying distances. In the imaging system, the input image is initially placed at $z = -f_1$, then passes through the acoustic meta-differentiator situated at $z = 0$, culminating in the output image located at the plane $z = f_2$. The acoustic meta-differentiator is engineered to operate at a frequency of 1 MHz, boasting a specific design with a radius of 30λ (45 mm) and a focal distance of $f = 40\lambda$ (60 mm). The amplitude object, symbolizing the numeral 2, possesses dimensions with a height of 6λ and a width of 3.75λ , and includes a slit measuring 0.75λ in width. When the object plane is shifted closer to the acoustic meta-differentiator, specifically at $f_1 = 36\lambda$, the resulting acoustic intensity distribution transmitting through the meta-differentiator along the propagation path is depicted in the left panel of Fig. R14a. In this scenario, the imaging plane is displaced to $f_2 = 45\lambda$, as determined by the lens imaging equation $2/f = 1/f_1 + 1/f_2$. Furthermore, the top-right and bottom-right panels in Fig. R14a exhibit the acoustic intensity and phase distributions extracted from the imaging plane, clearly illustrating that all edges of the numeral 2 are prominently highlighted

Figure R14: Imaging performance of acoustic meta-differentiator when the object placed at various positions in the imaging system. (a) Acoustic intensity distribution after transmitting through the meta-differentiator along the propagation path with the object placed at $f_1 = 36\lambda$ (left panel). Acoustic intensity (top right panel) and phase (bottom right panel) distributions extracted from the imaging plane at $f_2 = 45\lambda$ as denoted by the yellow dashed line. (b) Equivalent results to (a), but for the acoustic field with the object and imaging planes residing at $f_1 = 40\lambda$ and $f_2 = 40\lambda$, respectively. (c) Same as (a), but for the object and imaging planes located at $f_1 = 44\lambda$ and $f_2 = 36.7\lambda$, respectively.

with nearly uniform intensity. We also explore the case where the object plane is precisely positioned at $f_1 = 40\lambda$, aligning with the focal position of the acoustic meta-differentiator. The corresponding acoustic intensity and phase profiles are presented in Fig. R14b. In this instance, the imaging plane resides at $f_2 = 40\lambda$, and the intensity profile at the imaging plane accurately captures all boundary information of the object, faithfully representing the shape of the numeral 2. Additionally, we have examined the scenario in which the object plane is situated at a greater distance from the acoustic meta-differentiator, specifically at $f_1 = 44\lambda$. The corresponding acoustic intensity and phase profiles are presented in Fig. R14c. It is evident that the imaging plane shifts to $f_2 = 36.7\lambda$, and the intensity profile at the imaging plane accurately captures all boundary information of the object, representing the shape of the numeral 2. Moreover, when the object is moved closer to the acoustic meta-differentiator, the resulting image undergoes proportional enlargement, in accordance with the imaging principles akin to those of optical lenses. In conclusion, our proposed meta-differentiator exhibits the capability to effectively detect objects positioned at varying distances, and the resultant enlargement or minification effects can be harnessed to observe minute objects in greater detail.

We have also discussed the constraints associated with the measurement configuration of our proposed structure. As elucidated previously, our composite meta-grating exhibits the capability to directly perform 2D spatial differentiation on objects, enabling real-time image processing for the detected objects. However, the current experimental setup is constrained by the field-scanning method utilizing a single hydrophone in conjunction with the translation stage. In this case, the Schlieren imaging method can be employed for real-time visualization of the acoustic field [*Phys. Rev. Appl.* 15, 054041 (2021); *Phys. Rev. Appl.* 20, 024015 (2023)], potentially capturing the dynamic motions of living organisms. Additionally, while our configuration proves effective for the edge detection of 2D objects, it encounters

challenges in imaging 3D objects. To address this limitation, integration with advanced imaging techniques such as ultrasonic computed tomography (UCT) or the time-of-flight (TOF) method is imperative to facilitate comprehensive 3D object measurement. Furthermore, our method theoretically accommodates operation in both transmission and reflective modes. However, the reflective mode necessitates the preservation of scatter waves while excluding incident waves, introducing intricate procedures in edge-enhanced imaging processing. The combination of ultrasonic pulses with reflective-mode imaging systems holds promise for improving the practical utility in biomedical imaging and medical diagnosis. Therefore, there is scope for further refining our method to enhance its applicability in practical biomedical engineering scenarios.

We have integrated the aforementioned discussions into the paper to elucidate the distance response of acoustic meta-differentiator on **Page 9**: “Furthermore, this 2D meta-differentiator demonstrates the capability to effectively detect objects positioned at different distances, and the resulting enlargement or minification effects can be leveraged to observe minute objects in greater detail (refer to Supplementary Note 11).”. We also have included Fig. R14 into the **Supplementary Note 11 (Figure S10)** accompanied by relevant discussions for a more comprehensive understanding of this aspect.

Moreover, we have incorporated additional comprehensive information pertaining to the limitations of measurement configuration on **Page 15**: “It is noteworthy that several limitations in the measurement configuration may impede the current applications of spatial differentiator. Primarily, the field-scanning method employing one single hydrophone together with translation stage is time-consuming. As an alternative, the Schlieren imaging method is promising for real-time visualization of acoustic fields [62], potentially capturing the dynamic motions of living organisms [7]. Additionally, our current configuration faces challenges in imaging 3D objects, and necessitates the imperative integration of contemporary imaging techniques to facilitate comprehensive 3D object measurement [14]. Furthermore, the reflective-mode imaging system combined with ultrasonic pulse method should be constructed for commoner applications in medical diagnosis [13]. Hence, there is an opportunity for further refinement of our methodology to enhance its applicability in practical biomedical engineering scenarios.”.

Comment P 3.8 — *References: In my opinion there is a significant issue in the bibliography as it does not include any reference about spiral diffraction grating structures. Further, the references list on acoustic metasurfaces should be expanded. Besides the reference cited above, many metasurfaces have been recently reported that realize Fresnel-type focusing structures.*

Response: We express our gratitude to the reviewer for highlighting this issue. Following the reviewer’s insightful suggestion, we have incorporated references related to spiral diffraction grating structures and acoustic metasurfaces into our manuscript. Furthermore, we have identified several pivotal papers pertaining to acoustic metasurfaces, particularly those involving Fresnel-type focusing structures. These references have been included and cited as Refs. [48-53,56,57] to enrich the academic context of our study.

Reviewer #1 (Remarks to the Author):

The authors have addressed my concerns and I support the manuscript for publication.

Reviewer #3 (Remarks to the Author):

The authors replied to all the referees' questions, improving the quality of the paper, especially concerning the characterization of the meta-differentiator performances, the novelty with respect to the state of the art, the bibliographic analysis.

The paper, together with the extensive additional notes, is now very detailed, exhaustive, and of a high-technical level.

Concerning the state of the art and the bibliography, the authors added many new references and described better the state of the art, highlighting the novel aspect of this work.

Please remove/change References [60] and [61], as they do not focus on lock-in but on pulse-compression.

So, although I have still some concerns about the practical use of the meta-differentiator in a real measurement setup, the article deserves publication here or perhaps in a more specific journal focused on applied physics, acoustics, etc.

At the moment, the use as a tool for an acoustic microscopy seems the preferred one, but new applications can open in the future.

Response to the Reviewers

We express our gratitude to all reviewers for their acceptance of our responses to their comments and for their invaluable contributions to the peer review process of this paper.

Response to Reviewer 1

Comment P 1.1 — *The authors have addressed my concerns and I support the manuscript for publication.*

Response: We thank the reviewer for thorough review of our responses and the endorsement of our paper for publication in this esteemed journal of high quality.

Response to Reviewer 2

Comment P 2.1 — *The authors replied to all the referees' questions, improving the quality of the paper, especially concerning the characterization of the meta-differentiator performances, the novelty with respect to the state of the art, the bibliographic analysis. The paper, together with the extensive additional notes, is now very detailed, exhaustive, and of a high-technical level. Concerning the state of the art and the bibliography, the authors added many new references and described better the state of the art, highlighting the novel aspect of this work. Please remove/changes References [60] and [61], as they do not focus on lock-in but on pulse-compression. So, although I have still some concerns about the practical use of the meta-differentiator in a real measurement setup, the article deserves publication here or perhaps in a more specific journal focused on applied physics, acoustics, etc. At the moment, the use as a tool for an acoustic microscopy seems the preferred one, but new applications can open in the future.*

Response: We would like to thank the reviewer for meticulous examination of our responses. We concur with the reviewer's observation regarding the predominant emphasis on pulse-compression within References [60] and [61], rather than on lock-in techniques. According to the reviewer's suggestion, we have removed References [60] and [61] from the manuscript to ensure precision and fidelity. In addition, we agree with the reviewer's assessment that our proposed methodology holds significant potential in the realm of acoustic microscopy. In the future study, we will be engaged in exploring additional applications and extensions of our method to further enhance its utility and applicability.